# TRPC3 and NALCN channels drive pacemaking in substantia nigra dopaminergic neurons

Ki Bum Um[1], Suyun Hahn[1], So Woon Kim[1], Yoon Je Lee[1], Lutz Birnbaumer[2], Hyun Jin Kim[1,3]*, Myoung Kyu Park[1,3]*

[1]Department of physiology, Sungkyunkwan University School of Medicine, Suwon, Republic of Korea; [2]Neurobiology Laboratory. National Institute of Environmental Health Sciences, North Carolina 27709, USA; and Institute of Biomedical Research (BIOMED), Catholic University of Argentina, Buenos Aires, Argentina; [3]Samsung Biomedical Research Institute, Samsung Medical Center, Seoul, Republic of Korea

**Abstract** Midbrain dopamine (DA) neurons are slow pacemakers that maintain extracellular DA levels. During the interspike intervals, subthreshold slow depolarization underlies autonomous pacemaking and determines its rate. However, the ion channels that determine slow depolarization are unknown. Here we show that TRPC3 and NALCN channels together form sustained inward currents responsible for the slow depolarization of nigral DA neurons. Specific TRPC3 channel blockade completely blocked DA neuron pacemaking, but the pacemaking activity in TRPC3 knock-out (KO) mice was perfectly normal, suggesting the presence of compensating ion channels. Blocking NALCN channels abolished pacemaking in both TRPC3 KO and wild-type mice. The NALCN current and mRNA and protein expression are increased in TRPC3 KO mice, indicating that NALCN compensates for TRPC3 currents. In normal conditions, TRPC3 and NALCN contribute equally to slow depolarization. Therefore, we conclude that TRPC3 and NALCN are two major leak channels that drive robust pacemaking in nigral DA neurons.

*For correspondence:
kimhyunjin@skku.edu (HJinK);
mkpark@skku.edu (MKyuP)

**Competing interest:** The authors declare that no competing interests exist.

## Introduction

Dopamine (DA) neurons in the substantia nigra pars compacta (SNc) are essential for controlling the motivational parts of brain functions such as voluntary movement, action selection, future movement, and reward-based learning (*Grace and Bunney, 1984*; *Schultz, 2007*; *da Silva et al., 2018*). Dysfunction of DA neurons is associated with many neuropsychiatric diseases including Parkinson's disease and schizophrenia (*Bozzi and Borrelli, 2006*). Midbrain DA neurons are slow pacemakers that continuously generate spontaneous action potentials, consequentially sustaining ambient DA levels in target areas including the striatum (*Grace et al., 2007*; *Morikawa and Paladini, 2011*). However, in response to unexpected reward stimuli, DA neurons also produce high-frequency burst discharges that accompany DA surges (*Schultz, 1998*; *Schultz, 2013*). Thus, the intrinsic basal activity of pacemaking is the basis on which DA neurons operate. The rate and regularity of DA neuron pacemaking are robust against many kinds of pharmacological and molecular perturbations including most known specific blockers for classical candidate pacemaker ion channels, such as HCN and voltage-activated $Ca^{2+}$ channels (VACCs; *Paladini et al., 2003*; *Kim et al., 2007*; *Guzman et al., 2009*). The robustness of pacemaking suggests the necessity and biological importance of basal DA levels and DA signaling in the brain (*Guzman et al., 2009*; *Surmeier et al., 2012*). Therefore, understanding the pacemaking processes of DA neurons is of paramount importance not only for basic physiological knowledge but also for pathological mechanisms useful in therapeutic approaches.

The ion channels underlying autonomous pacemaking, therefore, have been the subject of investigation for more than three decades. The key feature of DA neuron pacemaking is the subthreshold slow depolarization during interspike intervals that requires a non-voltage-dependent background sodium current (*Khaliq and Bean, 2010*). At present, the dominant view is that, in SNc DA neurons, robust pacemaking depends on leak-like nonselective cation channels that constitutively open at relatively hyperpolarized membrane potentials. So far, the pacemaking of DA neurons is only stopped by SKF-96365 and 2-aminoethoxydiphenyl borate (2-APB), nonspecific blockers for nonselective cation channels (*Kim et al., 2007*) that are also blockers for transient receptor potential-canonical (TRPC) channels (*Nilius and Flockerzi, 2014*; *Lievremont et al., 2005*). TRPC channels constitute a group of nonselective cation channels of the transient receptor potential (TRP) superfamily. The TRPC members can be further divided into four subgroups (TRPC1, TRPC2, TRPC4/5, and TRPC3/6/7) and form homotetramers and heterotetramers among TRPCs and other types of TRP proteins (*Wang et al., 2020*). Among several members of the TRPC family, TRPC3 may be a potential candidate for pacemaker channels in DA neurons because of its constitutive voltage-independent activities at very low membrane potentials (*Dietrich et al., 2003*; *Zhou et al., 2008*).

Recently, NALCN has been identified as the background Na$^+$ leak channel that determines resting Na$^+$ permeability and neuronal excitability in the central nervous system (*Lu et al., 2007*). There is increasing evidence that NALCN contributes to the resting membrane potentials and endogenous activities in several neurons (*Lu et al., 2007*; *Lu et al., 2009*; *Shi et al., 2016*; *Lutas et al., 2016*; *Yeh et al., 2017*). NALCN is a Na$^+$-permeable nonselective cation channel and, therefore, can be a strong candidate for pacemaker channels in DA neurons. In the SNc DA neuron of conditional NALCN knockout (KO) mice, spontaneous firing activity was severely decreased in most cells (*Philippart and Khaliq, 2018*), implying that NALCN may be an essential channel for pacemaking in SNc DA neurons.

In this study, we use TRPC3 KO mice, specific TRPC3 channel blockers (*Kiyonaka et al., 2009*; *Schleifer et al., 2012*), and NALCN channel blockers that we recently identified (*Hahn et al., 2020*) to report that TRPC3 and NALCN are two major leak channels essential for robust pacemaking in SNc DA neurons.

## Results

### Blockade of either nonselective cation channels or TRPC3 channels stops pacemaking of SNc DA neurons

Endogenous firing activity of SNc DA neurons was measured by whole-cell patch-clamp recording in midbrain slices of TH-eGFP transgenic mice in which DA neurons express eGFP (*Figure 1A*, top; *Jang et al., 2015*). The recorded DA neuron was visualized by Alexa-594 dye in patch pipette and/or presented as a 3D reconstructed image (*Figure 1*). Most SNc DA neurons recorded from midbrain slices exhibited a very regular firing rhythm with an average firing rate of 3.28 ± 0.13 Hz (*Figure 3—figure supplement 1*, n = 17). DA neuron pacemaking can be characterized by slow depolarization during the interspike interval, which is further divided into three phases: I, the initial afterhyperpolarization due to the Ca$^{2+}$ overload and SK channel activation (*Nedergaard et al., 1993*); II, the middle, very long, steady, and slow depolarization; and III, the last accelerated depolarization (*Figure 1B*). Phase II, occupying more than 3/4 of the interspike interval (*Figure 1C*), most clearly denotes slow depolarization (*Figure 1B*, green dotted line). Slow depolarization is absolutely necessary for pacemaking (*Nedergaard et al., 1993*; *Grace and Onn, 1989*; *Ping and Shepard, 1996*) and determines the pacemaking rate (*Figure 1D*). A mild and linear slope in the slow depolarization (*Figure 1B*) implies the presence of small but continuous depolarizing currents during interspike intervals. HCN and VACCs have been at the center of debate in pacemaking mechanisms of DA neurons (*Chan et al., 2007*; *Branch et al., 2014*). Isradipine blocks dihydropyridine (DHP)-sensitive Ca$^{2+}$ channels including Ca$_V$1.3 channels, a dominant type of VACCs in DA neurons (*Guzman et al., 2009*), and ZD-7288 is an HCN channel blocker (*Chan et al., 2007*; *Zolles et al., 2006*). As previously reported (*Kim et al., 2007*; *Guzman et al., 2009*), they do not affect spontaneous firing rate at all (*Figure 1—figure supplement 1A,B*), indicating that L-type Ca$^{2+}$ and HCN channels are not essential for pacemaking. In addition, the co-application of isradipine and ZD-7288 also failed to slow down the spontaneous firing rate in SNc DA neurons (*Figure 1—figure supplement 1A,B*), indicating that these two channels have little importance in the pacemaking of DA neurons. Nevertheless, during interspike intervals, isradipine

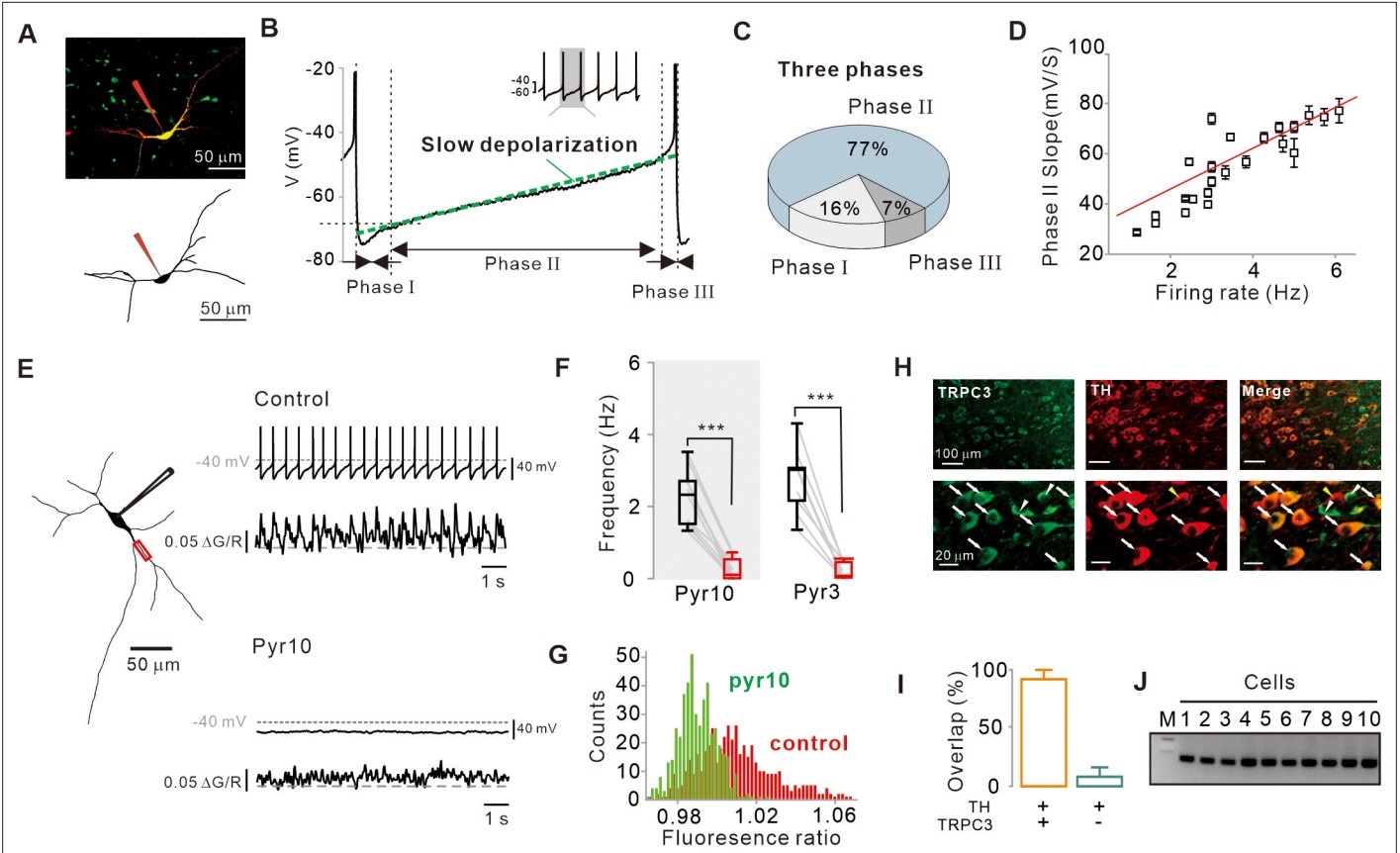

**Figure 1.** Importance of subthreshold slow depolarization and TRPC3 channels in the pacemaking of SNc DA neurons. (**A**) Projection (top) and 3D reconstruction (bottom) images of TH-positive dopamine neurons from SNc slices filled with Alexa-594 and Fluo-4 through a whole-cell patch pipette. (**B**) Representative voltage trajectory during the inter-spike interval reveals three phases of slow depolarization for pacemaking of SNc dopamine neurons in midbrain slices. Inset, a longer period of pacemaking in the same neurons. (**C**) Pie chart showing the duration of each phase in slow depolarization (n = 20 from 12 mice). (**D**) Slope of slow depolarization (determined as in (**B**), green line) plotted against firing frequency (n = 23 from 15 mice). Red line is the best fitting line by linear regression. (**E**) Whole-cell recording from dopamine neurons in midbrain slices (left) and Ca$^{2+}$ imaging at dendritic locations (left, red rectangle). Pyr10 (50–100 µM) inhibited both spontaneous firing (right, top) and dendritic Ca$^{2+}$ oscillations (right, bottom). Representative traces were obtained from the same neuron. (**F**) Summary plot showing inhibition of firing frequency by pyr10 (n = 11 from 7 mice, ***p<0.0001) and pyr3 (n = 9 from 5 mice, ***p<0.0001). Black: control, red: after drug application. (**G**) All-points histogram of Ca$^{2+}$ fluorescence shows that the fluorescence oscillation was reduced by pyr10. (**H**) Double immunofluorescence staining images for TRPC3 (left, green), TH (middle, red), and merge (right) from the SNc. Arrows indicate co-expression of TRPC3 and TH, while arrowheads indicate cells expressing only TRPC3 without TH. (**I**) Histogram for co-expression of TRPC3 and TH in SNc neurons (from three mice). (**J**) TRPC3 RT-PCR profiles from single dopamine neurons. All statistical data were analyzed by one-way ANOVA.

The online version of this article includes the following source data and figure supplement(s) for figure 1:

**Source data 1.** Source data of each group in *Figure 1*.

**Figure supplement 1.** Pharmacological examination of potential ion channels affecting the pacemaking activities of nigral dopamine neurons.

**Figure supplement 1—source data 1.** Source data of each group in *Figure 1—figure supplement 1*.

**Figure supplement 2.** Blockade of L-type Ca$^{2+}$ and HCN channels does not stop the pacemaking of acutely dissociated SNc dopamine neurons.

**Figure supplement 2—source data 1.** Source data of each group in *Figure 1—figure supplement 2*.

---

weakly reduced the afterhyperpolarization possibly via reduced SK channel currents by decreasing Ca$^{2+}$ influx (*Poetschke et al., 2015*) and slightly reduced the later slow and accelerated depolarization (*Figure 1—figure supplement 1E,G*), collectively resulting in no significant effect on spontaneous firing rate. On the other hand, ZD-7288 increased slightly afterhyperpolarization (*Figure 1—figure supplement 1F,G*), but the increase was too small to affect the firing rate. Regarding the pharmacological effects on firing inhibition, we obtained the same results from the cell-attached patch-clamp recordings in the freshly dissociated single SNc DA neurons (*Figure 1—figure supplement*

*2*), indicating that our results from the midbrain slices are not related to either incomplete channel antagonism in brain slice experiments (*Guzman et al., 2009*) or perturbation of the intracellular milieu by whole-cell pipette solution (*Zolles et al., 2006*).

Previously, using freshly dissociated SNc DA neurons, we reported that nonselective cation channels (NSCCs) are essential for pacemaking (*Kim et al., 2007*). Consistent with this report, nonspecific TRPC channel blockers, such as SKF-96365 and 2-ABP, strongly inhibited the spontaneous firing of DA neurons in the midbrain slices (*Figure 1—figure supplement 1C,D*). TRPC channels constitute a large and functionally versatile family of nonselective cation channel proteins (*Gees et al., 2010*). Among many TRPC members, TRPC3 is known to be highly expressed in the brain (*Sylvester et al., 2001*; *Clapham, 2003*; *Clapham et al., 2005*) and a constitutively active ion channel (*Dietrich et al., 2003*; *Zhou et al., 2008*). Therefore, we used pyrazole derivatives such as pyr3 and pyr10 to selectively inhibit TRPC3 channels, although pyr10 is reported to partially inhibit TRPC6, too (*Kiyonaka et al., 2009*; *Schleifer et al., 2012*). These channel blockers completely abolished spontaneous firing of DA neurons in the midbrain slices (*Figure 1E,F*), together with the disappearance of dendritic $Ca^{2+}$ oscillations, which were measured by Fluo-4 in the whole-cell patch pipette (*Figure 1E,G*). Double immunofluorescence staining of the midbrain slices containing the SNc (*Figure 1H,I*; *Clapham et al., 2005*) showed that TH-positive DA neurons were mostly overlapped with TRPC3-positive neurons (overlaps = 92%). In addition, single-cell RT-PCR from the isolated SNc DA neurons showed expression of TRPC3 mRNA in all cells examined (*Figure 1J*). Taken together, these results raise the possibility that TRPC3 is a nonselective cation channel essential for the pacemaking of SNc DA neurons.

## TRPC3 encodes a nonselective cation channel essential for the slow depolarization of SNc DA neurons during interspike intervals

In SNc DA neurons of midbrain slices, pyr10 eliminated slow depolarization and spontaneous firing completely (*Figures 1E and 2A*). Under this condition, the injection of a small amount of continuous current, which resembles a leak current, revived spontaneous firing, at a rate that correlated with an injected current size (*Figure 2*). When the firing rate was resuscitated to the control level, voltage traces of the regenerated slow depolarization and action potential were completely aligned with those before pyr10 treatment (*Figure 2A*, right bottom; *Figure 2—figure supplement 1*), indicating that the channel inhibited by pyr10 could be a leak-like channel. Consistent with these data, the firing rate was gradually increased within the pacemaking range by a slow ramp-like increase in current injection (*Figure 2*, right-top), implying that the amount of leak current determines the pacemaking rate in SNc DA neurons.

Next, we examined whether the blockade of the TRPC3 channel affects subthreshold membrane potentials of SNc DA neurons. In spontaneously firing DA neurons, application of tetrodotoxin (TTX), a voltage-dependent $Na^+$ channel blocker, completely suppressed spontaneous action potentials but slow oscillatory potentials (SOPs) survived in a slightly more depolarized state (*Figure 2*), as previously reported (*Nedergaard et al., 1993*; *Estep et al., 2016*). These SOPs are not essential for pacemaking in DA neurons (*Guzman et al., 2009*), but are mediated by cyclic interactions between L-type $Ca^{2+}$ channels and $Ca^{2+}$-dependent SK channels within an adequate range of membrane potentials (*Ping and Shepard, 1996*; *Wilson and Callaway, 2000*). Consistent with these reports, further treatment with isradipine suppressed SOPs and lowered the membrane potential (*Figure 2*). In contrast, pyr10 not only suppressed SOPs but also hyperpolarized membrane potentials more strongly than isradipine (*Figure 2D,E*), indicating that pyr10-inhibited channels are able to depolarize the membrane potential even in the more hyperpolarized state. After pyr10 hyperpolarized the membrane potential, the total replacement of external $Na^+$ with equimolar large cation N-methyl-d-glucamine (NMDG) further hyperpolarized the membrane potential (*Figure 2E*), at a value similar to that predicted by the Goldman-Hodgkin-Katz (GHK) equation ($V_{GHK}$ = –72.57 mV; see Materials and methods details). All these data suggest that TRPC3 substantially contributes to slow depolarization of the membrane potential in DA neurons, but not alone.

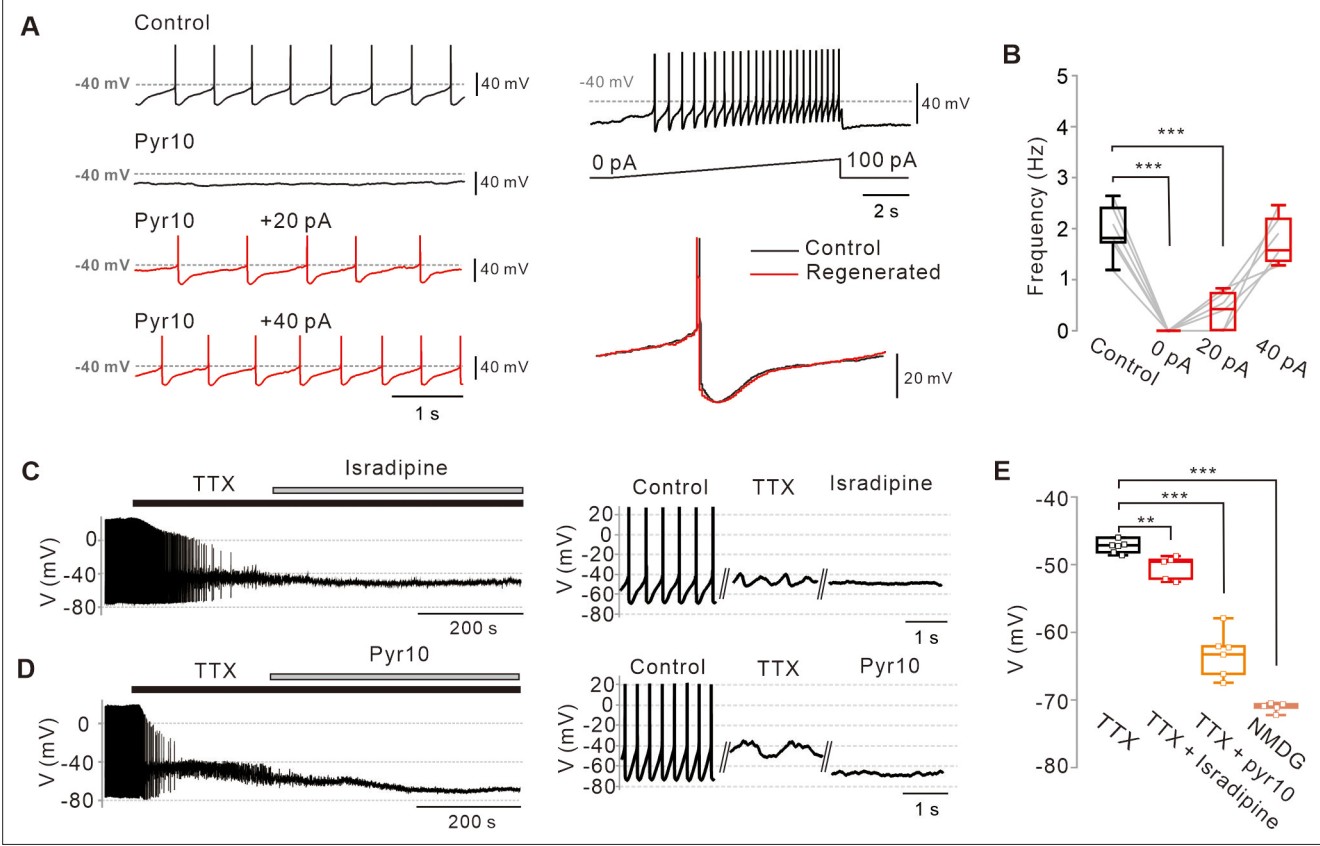

**Figure 2.** Abolition of pacemaking activity by TRPC3 blockade and its resumption by compensating leak-like current injection. (**A**) Voltage traces from SNc dopamine neurons in the midbrain slices (left). Pyr10 (left, black trace, 50 μM) completely inhibited spontaneous firing, but it was rescued by somatic linear current injection (left, red trace). Representative voltage traces were obtained from the same neuron. Pacemaking activities under the presence of pyr10 were gradually revived by slow ramp-current injection (right upper). No significant shape changes between control and revived action potentials (APS, right bottom). (**B**) Box plots for pacemaking frequencies from data in a (n = 6 from 3 mice). ***p<0.0001 for control versus 0 pA; ***p<0.001 for control versus +20 pA; p>0.5 for control versus +40 pA. (**C**) Changes of membrane potentials in SNc dopamine neurons by L-type calcium channel blockade after TTX (0.5 μM) treatment (isradipine 5 μM). (**D**) Application of pyr10 (50 μM) hyperpolarized the membrane potential in the presence of TTX. (**E**) Summary of membrane potential changes by TTX (n = 7 from 6 mice), TTX and isradipine (n = 5 from 3 mice), and TTX and pyr10 (n = 6 from 3 mice). **p<0.01 for TTX versus TTX and isradipine; ***p<0.001 for TTX versus TTX and pyr10; ***p<0.001 for TTX versus NMDG. All statistical data were analyzed by one-way ANOVA.

The online version of this article includes the following source data and figure supplement(s) for figure 2:

**Source data 1.** Source data of each group in *Figure 2*.

**Figure supplement 1.** Comparisons of time courses of the action potentials between the control and regenerated firings under the inhibition of TRPC3 channels.

**Figure supplement 1—source data 1.** Source data of each group in *Figure 2—figure supplement 1*.

## Selective TRPC3 channel antagonism of Pyr10 in SNc DA neurons in WT mice and compensation of TRPC3-induced leak current in TRPC3 KO mice

To overcome the limitations of pharmacological approaches, we used TRPC3 KO mice (*Figure 3*) in which the recorded SNc DA neurons with the whole-cell patch pipette containing neurobiotin were later confirmed by double immunostaining of TH and streptavidin after the experiments (*Figure 3A*, neurobiotin, green; TH, red). Surprisingly, the spontaneous firing of DA neurons in TRPC3 KO mice survived, and the spontaneous firing rate did not differ from that of wild-type (WT) littermates (*Figure 3—figure supplement 1*, n = 23, p=0.23). Moreover, we could not find any significant difference in many electrical properties of DA neurons between TRPC3 KO and WT mice (*Figure 3—figure supplement 1A–I*), suggesting that the DA neuron pacemaking is perfectly normal in TRPC3 KO mice.

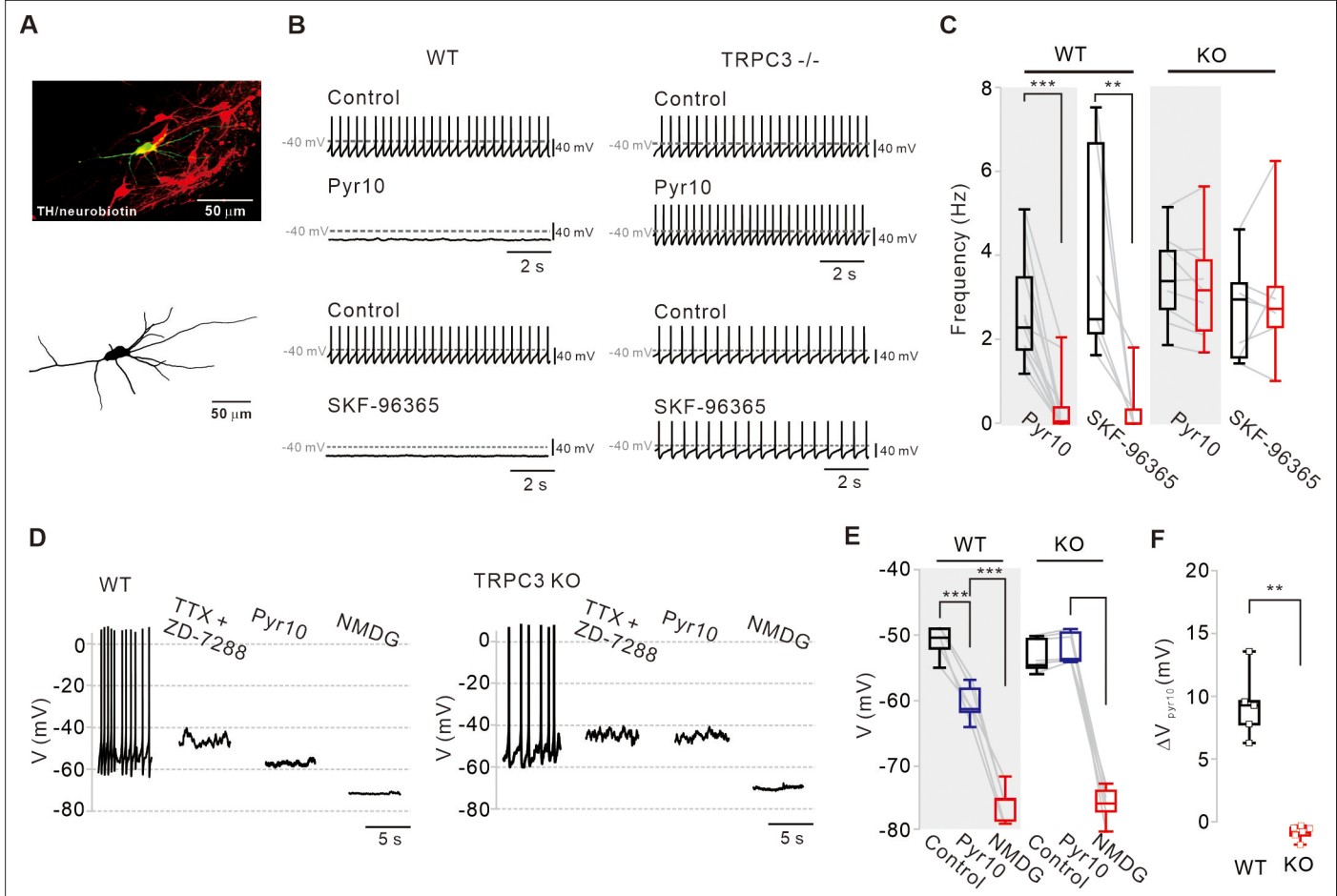

**Figure 3.** Different effects of TRPC blockers on the pacemaking activities of DA neurons in wild-type and TRPC3 KO mice. (**A**) Dopamine neurons were identified from SNc slices by post hoc staining (top) for TH (red) and neurobiotin (green) in TRPC3 knockout (KO) and wild-type (WT) mice. The lower image is a 3D reconstructed dopamine neuron that was previously recorded with a patch pipette (bottom). (**B**) Application of pyr10 (50 µM) or SKF-96365 (20 µM) inhibited the pacemaking of dopamine neurons in midbrain slices from WT (left) mice, but not from TRPC3 KO mice (right). Traces showing the effect of each blocker were obtained from the same neuron. (**C**) Summary plots for the effects of pyr10 or SKF-96365 on the spontaneous firing frequency from WT (pyr10, n = 13 from 11 mice, ***p<0.001; SKF-96365, n = 6 from 6 mice, **p<0.01) and TRPC3 KO (pyr10, n = 9 from 6 mice, p>0.05; SKF-96365, n = 7 from 4 mice, p>0.05) mice (black, control; red, after drug application). (**D**) Different effects of pyr10 (10 µM) on the membrane potentials in the presence of TTX (0.5 µM) and ZD-7288 (20 µM), measured in acutely dissociated dopamine neurons from WT (left) and TRPC3 KO (right) mice. Representative voltage traces were obtained from the same neuron. (**E**) Summary of membrane potential changes by pyr10 in the presence of TTX and ZD-7288 in WT (pyr10, n = 5 from 5 mice, ***p<0.001; NMDG, ***p<0.001) and TRPC3 KO (pyr10, n = 6 from 5 mice, p=0.57; NMDG, ***p<0.001). (**F**) Box plots for membrane potential changes by pyr10 (ΔV$_{pyr10}$) in WT and TRPC3 KO mice. Data from (**E**) (**p<0.01). All statistical data were analyzed by one-way ANOVA.

The online version of this article includes the following source data and figure supplement(s) for figure 3:

**Source data 1.** Source data of each group in *Figure 3*.

**Figure supplement 1.** Comparison of electrophysiological properties of DA neurons between TH-GFP, TRPC3 WT, and TRPC3 KO mice.

**Figure supplement 1—source data 1.** Source data of each group in *Figure 3—figure supplement 1*.

**Figure supplement 2.** Different effects of TRPC channel blockers on the pacemaking of SNc DA neurons between TRPC3 KO and WT mice.

**Figure supplement 2—source data 1.** Source data of each group in *Figure 3—figure supplement 2*.

**Figure supplement 3.** Pharmacological examinations of potential ion channels compensating for the pacemaking of DA neurons in TRPC3 KO mice.

**Figure supplement 3—source data 1.** Source data of each group in *Figure 3—figure supplement 3*.

**Figure supplement 4.** Replacement of extracellular Na$^+$ with NMDG decreases intracellular [Ca$^{2+}$]$_c$ levels and does not activate SK channels in SNc DA neurons.

**Figure supplement 4—source data 1.** Source data of each group in *Figure 3—figure supplement 4*.

Therefore, we verified whether selective TRPC3 channel blockers work in these mice. Interestingly, in the TRPC3 KO mice, the application of pyr10 did not affect the spontaneous firing rate of DA neurons at all (*Figure 3B,C*), whereas pyr10 completely blocked the pacemaking of DA neurons in the WT mice (*Figure 3*). These data strongly indicate that the inhibitory action of pyr10 on DA neuron pacemaking in WT mice should be mediated by selective TRPC3 channel antagonism. In addition, unlike WT mice, SKF-96365, which blocks all members of TRPC channels (*Nilius and Flockerzi, 2014*; *Hahn et al., 2020*), did not affect the pacemaking rate of DA neurons in TRPC3 KO mice (*Figure 3B,C*). The same results were obtained from acutely isolated SNc DA neurons (*Figure 3—figure supplement 2*). Therefore, we next examined whether pyr10 affects the subthreshold membrane potentials of DA neurons in these mice. To measure the membrane potentials more exactly, we pretreated DA neurons with TTX and ZD-7288 to remove interferences from action potential-induced membrane potential fluctuations and hyperpolarization-activated currents by HCN channels, respectively. Under this condition, pyr10 hyperpolarized the membrane potential of DA neurons in WT mice as expected (*Figure 3D*, left), but not in TRPC3 KO mice (*Figure 3D*, right). When all extracellular $Na^+$ was replaced by equimolar NMDG after treatment with pyr10, the membrane potential of DA neurons was maximally hyperpolarized (*Figure 3E,F*). In the absence of extracellular $Na^+$, $Na^+/Ca^{2+}$ exchangers may activate SK channels by increasing intracellular $Ca^{2+}$ concentration ($[Ca^{2+}]_c$) and then affect the membrane potential that we measured. Therefore, we examined whether the SK channel blocker apamin (100 nM) affects the membrane potential when extracellular $Na^+$ was replaced with NMDG. However, there was no change in the membrane potentials, suggesting that this was not the case (*Figure 3—figure supplement 4*). These data suggest that the sustained inward currents produced by TRPC3 channels must be compensated by other $Na^+$-permeable ion channels in TRPC3 KO mice.

In TRPC3 KO mice, the input resistance of DA neurons did not differ from that in WT mice (*Figure 3—figure supplement 1I*) and SKF-96365 did not affect the spontaneous firing rate in the midbrain slices (*Figure 3C*) and in freshly dissociated DA neurons (*Figure 3—figure supplement 2*). In TRPC3 KO mice, another nonspecific but more complex TRPC channel blocker, 2-APB, also failed to stop pacemaking (*Figure 3—figure supplement 3A,B*), but unlike SKF-96365, it slowed down the spontaneous firing rate, possibly, by complex actions to other ion channels (*Nilius and Flockerzi, 2014*; *Lievremont et al., 2005*). In addition, ZD-7288 and isradipine had no significant effect on the spontaneous firing rate of DA neurons in TRPC3 KO mice (*Figure 3—figure supplement 3A,B*), suggesting that L-type $Ca^{2+}$ and HCN channels do not compensate for pacemaking in these mice. Therefore, it is highly likely that the leak-like inward current induced by TRPC3 channels appears to be completely compensated by SKF-96365-resistant and $Na^+$-permeable ion channels in TRPC3 KO mice.

## NALCN is another non-selective cation channel essential for the slow depolarization of SNc DA neurons

Recently, the $Na^+$ leak channel NALCN has been reported to contribute to resting $Na^+$ permeability and basal excitability in neurons (*Lu et al., 2007*; *Ren, 2011*). However, the neonatal lethality of NALCN KO mice and lack of specific blockers (*Lu et al., 2007*; *Lu et al., 2009*; *Shi et al., 2016*; *Lutas et al., 2016*; *Yeh et al., 2017*) have hampered investigation of the potential role of NALCN in the pacemaking of DA neurons. Recently, we have found that N-benzhydryl quinuclidine (NBQN) compounds, including L-703,606, are a potent blocker for NALCN channels without affecting TRPC channels (*Hahn et al., 2020*). Therefore, we applied L-703,606 to SNc DA neurons in midbrain slices and observed the complete abolition of pacemaking (*Figure 4A*). Under this condition, step-by-step linear current injections by whole-cell patch pipette to the neuron restored regular pacemaking activity (*Figure 4B,D*), which is similar to in TRPC3 channels (*Figure 2*). When the firing rate was resuscitated to the control level, the voltage traces between the revived firings and those before L-703,606 treatment were completely aligned to each other (*Figure 4C*, *Figure 4—figure supplement 2*), indicating that the pacemaking can be completely revived by a leak-like current. The abolition of spontaneous firing by L-707,606 was also confirmed by cell-attached patch-clamp recording in freshly dissociated single SNc DA neurons (*Figure 4—figure supplement 1*). Next, we examined how much L-703,606 affects the subthreshold membrane potential of DA neurons in TRPC3 KO and WT mice. After pretreatment with TTX and ZD-7288, the application of L-703,606 hyperpolarized the membrane potential to the same extent as TRPC3 blockers in the acutely dissociated DA neurons of WT mice but resulted in greater hyperpolarization of the membrane potential in TRPC3 KO mice (*Figure 4E–G*). The replacement of

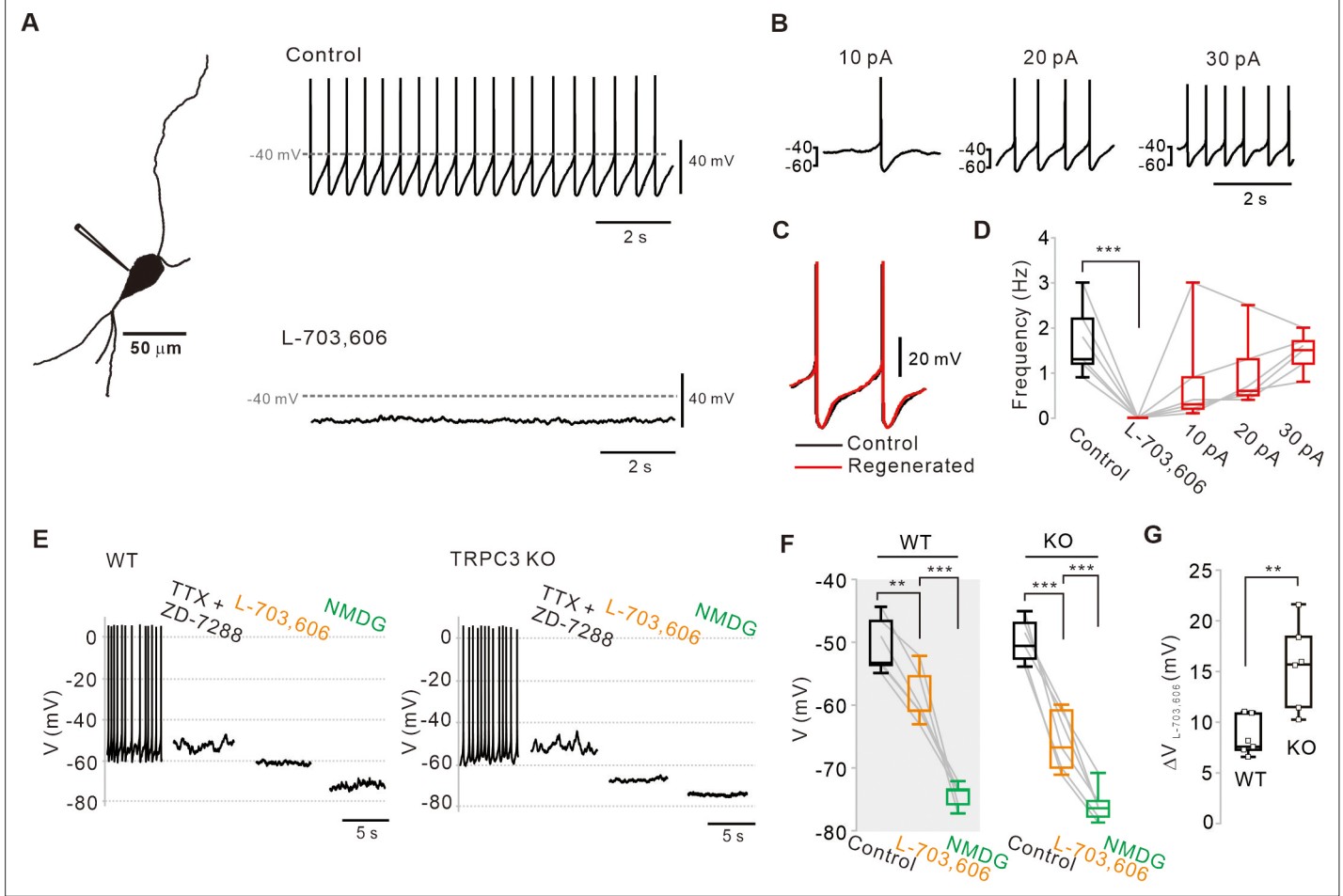

**Figure 4.** Abolition of pacemaking activity by NALCN channel blockers and its resumption by compensating leak-like current injection. (**A**) A 3D reconstruction image of an SNc dopamine neuron in a midbrain slice with a whole-cell patch pipette. Application of L-703,606 (10 μM) inhibited spontaneous firing. Representative voltage traces were obtained from the same neuron. (**B**) Spontaneous firing in the presence of L-703,606 was gradually rescued by somatic linear current injection (n = 6 from 4 mice). (**C**) Alignment of AP waveforms (normalized with time) between control (black) and regenerated (red, +30 pA). No significant changes in the shapes between control and revived APs. (**D**) Box plots for pacemaking frequencies before and after L-703,606 treatment and during somatic current injections in the presence of L-703,606 in SNc dopamine neurons (n = 6 from 4 mice). ***p<0.001 for control versus L-703,606; p>0.1 for control versus +10, +20, and +30 pA. (**E**) Representative traces for membrane potential changes in acutely dissociated dopamine neurons by L703,606 (5 μM) in the presence of TTX (0.5 μM) and ZD-7288 (20 μM) between wild-type (left, n = 6 from 4 mice) and TRPC3 KO mice (right, n = 6 from 4 mice). (**F**) Summary of membrane potential changes by L-703,606 in the presence of TTX and ZD-7288 in WT and KO. **p>0.01 for TTX and ZD-7288 versus L-703,606 from WT; ***p<0.001 for L-703,606 versus NMDG from WT; ***p<0.001 for TTX and ZD-7288 versus L-706,606 from TRPC3 KO; ***p<0.001 for L-703,606 versus NMDG from TRPC3 KO. (**G**) Summary plots for voltage differences (ΔV$_{L-703,606}$) changed by L-703,606 between WT and KO mice (**p<0.01). All statistical data were analyzed by one-way ANOVA.

The online version of this article includes the following source data and figure supplement(s) for figure 4:

**Source data 1.** Source data of each group in *Figure 4*.

**Figure supplement 1.** NALCN channel blockers stopped the pacemaking of acutely dissociated SNc DA neurons.

**Figure supplement 1—source data 1.** Source data of each group in *Figure 4—figure supplement 1*.

**Figure supplement 2.** Comparisons of time courses of the action potentials between the control and regenerated firings under the inhibition of NALCN channels.

**Figure supplement 2—source data 1.** Source data of each group in *Figure 4—figure supplement 2*.

external Na$^+$ with equimolar NMDG maximally hyperpolarized the membrane potentials in SNc DA neurons of both mice to the same extent (*Figure 4E,F*). All these data indicate that NALCN could be another nonselective cation channel essential for the pacemaking of SNc DA neurons. In TRPC3 KO mice, NALCN channels appear to compensate the TRPC3-mediated sustained depolarizing current essential for pacemaking in SNc DA neurons.

## Enhancement of NALCN currents and mRNA and protein expression of SNc DA neurons in TRPC3 KO mice

NALCN is a G-protein-coupled receptor channel that can be activated by substance P and neurotensin (NT; *Lu et al., 2009*; *Hahn et al., 2020*). Therefore, to examine more clearly whether NALCN currents are increased in TRPC3 KO mice, we measured NT-evoked NALCN currents in freshly dissociated SNc DA neurons directly. Because of the rapid desensitization of NT-evoked

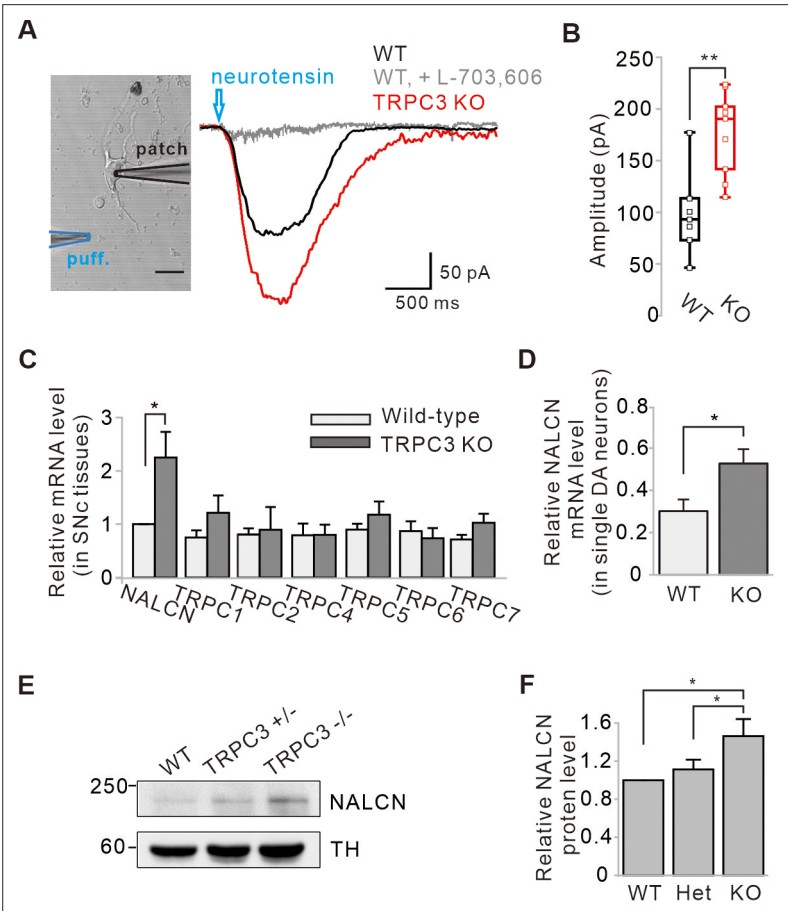

**Figure 5.** Enhancement of NALCN currents and mRNA and protein expressions in SNc dopamine neurons of TRPC3 KO mice. (**A**) An acutely dissociated dopamine neuron from the SNc was whole-cell patched and neurotensin (NT, 10 μM) was applied to dendritic compartments by a micro-puff system (left, blue, duration = 1). The NT-evoked NALCN currents were larger in the TRPC3 KO mice (red, n = 9 from 4 mice) than in the wild-type mice (black, n = 7 from 5 mice). Holding potential = –60 mV. (**B**) Summary of the current amplitudes evoked by NT in WT and TRPC3 KO (KO). p=0.0019 for WT versus TRPC3 KO. (**C**) Bar graphs showing the relative mRNA levels of NALCN and TRPC channels in SNc tissues from WT and TRPC3 KO mice (n = 5 mice). *p<0.05 for wild type versus TRPC3 KO from NALCN. (**D**) Relative NALCN mRNA levels of single SNc dopamine neurons between WT (n = 36 from 4 mice) and TRPC3 KO mice (n = 37 from 4 mice, *p<0.05). (**E**) Immunoblotting of TH (top) and NALCN (bottom) showing expression levels of NALCN protein in SNc tissues of wild-type, TRPC3 hetero (+/−) and KO (−/−) mice. (**F**) Comparisons of expression levels of NALCN protein in SNc tissues of wild-type, TRPC3 hetero and KO mice (n = 3, *p<0.05). All statistical data were analyzed by one-way ANOVA.

The online version of this article includes the following source data for figure 5:

**Source data 1.** Source data of each group in *Figure 5*.

NALCN currents in normal DA neurons (*Hahn et al., 2020*), we used a micropressure puff system (10 µM, 1 s). The NT-evoked NALCN currents in dissociated DA neurons from TRPC KO mice were significantly larger than those in WT mice (*Figure 5A,B*), demonstrating that the NALCN currents are increased in TRPC3 KO mice.

Next, we examined the expression of NALCN mRNA levels in both SNc tissues and dissociated single DA neurons from TRPC3 KO and WT mice using quantitative RT-PCR (qRT-PCR) (*Figure 5C,D*). In both cases, the expression of NALCN mRNAs in TRPC3 KO mice was significantly higher than in WT mice. However, mRNAs of the other members of TRPC channels in TRPC3 KO mice were not changed (*Figure 5C*). This is in line with our previous conclusion that NALCN channels compensate for the leak currents in TRPC3 KO mice rather than other members of TRPC. Furthermore, using western blot analysis, we confirmed that NALCN proteins were more significantly increased in heterozygous and homozygous TRPC3 KO mice than in WT littermates (*Figure 5E,F*).

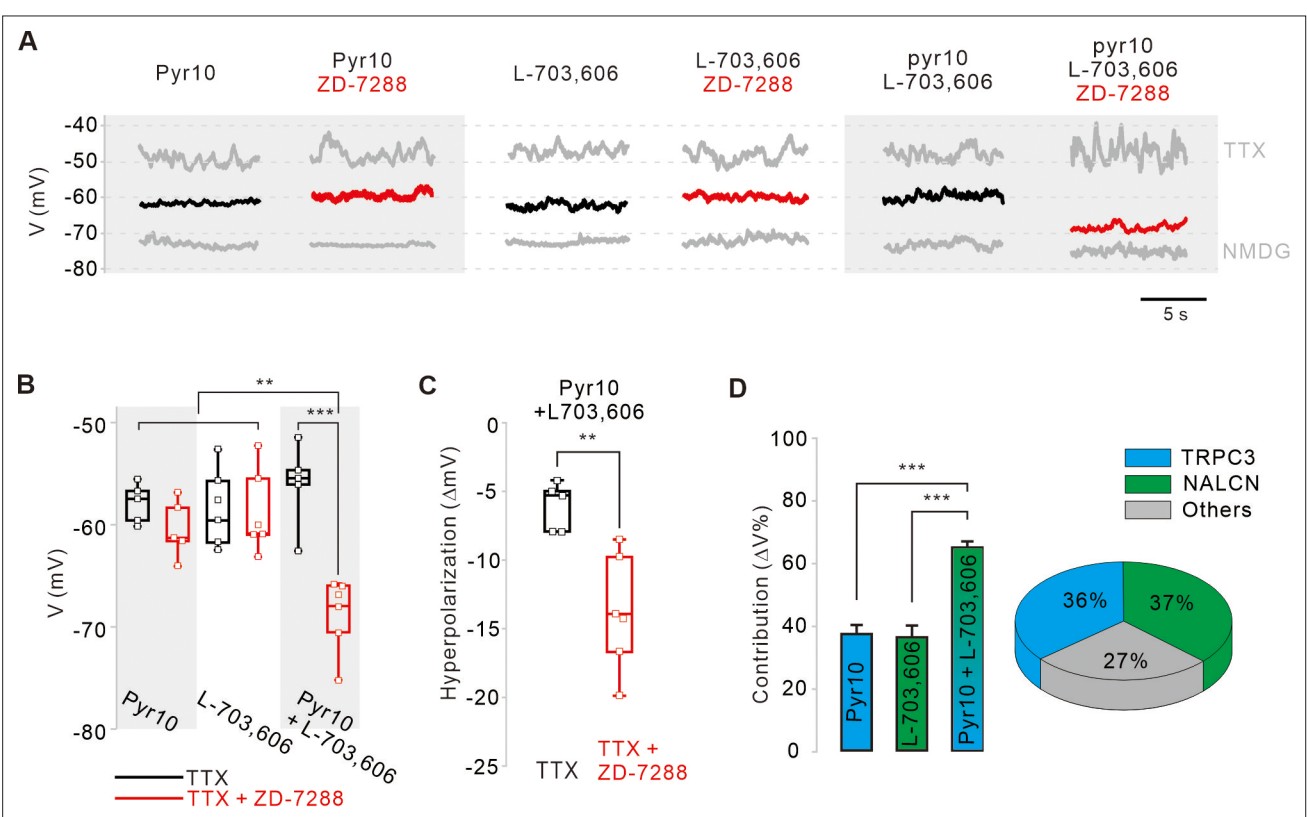

**Figure 6.** Relative contributions of TRPC3, NALCN, and HCN channels to the subthreshold depolarization of membrane potentials in SNc DA neurons. (**A**) Steady-state membrane potentials after treatment of acutely dissociated SNc DA neurons with TTX (0.5 µM) were measured by whole-cell patch-clamp recording. Relative changes of the membrane potentials in the silenced DA neurons by application of pyr10 (10 µM), L-703,606 (5 µM), and both blockers together were compared with or without ZD-7288. The maximally hyperpolarized membrane potentials were measured by the replacement of extracellular Na$^+$ with NMDG. (**B**) Summaries of the membrane potential changes by pyr10, L-703,606, and both. TTX alone (black, pyr10, n = 5 from 5 mice; L-703,606, n = 6 from 5 mice; pyr10 and L-703,606, n = 5 from 3 mice). TTX and ZD-7288 (red, pyr10, n = 5 from 6 mice; L-703,606, n = 6 from 4 mice; pyr10 and L-703,606, n = 6 from 3 mice). ***p<0.001 for TTX versus TTX and ZD-7288 from pyr10 and L-703,606; **p<0.01 for pyr10 and L-703,606 from TTX and ZD-7288 versus all others. (**C**) Box plots for the hyperpolarization of the membrane potential induced by ZD-7288 after blocking both TRPC3 and NALCN channels. Pyr10 and L-703,606 (black, n = 5 from 3 mice), ZD-7288 (red, TTX and ZD-7288, n = 6 from 3 mice, **p<0.05). (**D**) Relative contribution of TRPC3 and NALCN to depolarization of the membrane potentials in the presence of TTX and ZD-7288. Relative contributions were calculated using the ratios of the degrees of hyperpolarization after blocker treatment and NMDG replacement ($\Delta V_{blocker}/\Delta V_{NMDG}$) from data **A**. p>0.8 for pyr10 versus L-703,606; ***p<0.001 for pyr10 versus pyr10 and L-703,606; ***p<0.001 for L-703,606 versus pyr10 and L-703,606. Pie chart (right) showing the relative contribution of TRPC3 (cyan) and NALCN (blue) to subthreshold depolarization of the membrane potential. All statistical data were analyzed using one-way ANOVA.

The online version of this article includes the following source data for figure 6:

**Source data 1.** Source data of each group in *Figure 6*.

## Equal contribution of TRPC3 and NALCN to subthreshold depolarization of membrane potentials in SNc DA neurons

Finally, we examined the relative contributions of the TRPC3 and NALCN channels to the subthreshold depolarization of membrane potentials in SNc DA neurons. Because DA neurons highly express hyperpolarization-activated HCN channels (*Branch et al., 2014*) and ZD-7288 had little effect on the peak of afterhyperpolarization (*Figure 1—figure supplement 1F*), we questioned whether HCN channels affect pyr10- or L-703,606-induced hyperpolarization of membrane potentials in SNc DA neurons. After silencing SNc DA neurons with TTX treatment, the application of pyr10 or L-703,606 hyperpolarized the membrane potential to the same extent, but the extent of hyperpolarization was not affected by ZD-7288 in both cases (*Figure 6A,B*). Ironically, when both pyr10 and L-703,606 were co-applied to SNc DA neurons without ZD-7288, the hyperpolarization was similar to those induced by a single treatment with each blocker (*Figure 6A,B*). However, in the presence of ZD-7288, the co-application of pyr10 and L-703,606 further hyperpolarized the membrane potential (*Figure 6*), suggesting that the blockade of these two channels together causes enough hyperpolarization to activate HCN channels in SNc DA neurons. Therefore, HCN channels appear to activate in response to sufficiently significant hyperpolarization of the membrane potential in SNc DA neurons. These results are consistent with previous data showing that HCN channels contribute very little to normal pace-making processes (*Figure 1—figure supplement 2E,G*). Analysis of the degrees of hyperpolarization induced by pyr10 and/or L-703,606 in the presence of ZD-7288 (*Figure 6D*) suggests that the TRPC3 and NALCN channels contribute equally to more than two-thirds of the leak conductance responsible for slow depolarization in SNc DA neurons.

## Discussion

The main finding from our combined electrical, molecular, immunohistochemical, and pharmacological experiments is that TRPC3 and NALCN channels are two major leak channels essential for the robust pacemaking of SNc DA neurons. Subthreshold slow depolarization during the interspike interval drives pacemaking of SNc DA neurons that basically generates regular spontaneous firing at 2–6 Hz (*Grace and Bunney, 1984*). SNc DA neurons strongly express TRPC3 and NALCN channels, which constitute a sustained leak-like inward current responsible for slow depolarization in SNc DA neurons. In addition, knocking out TRPC3 channels increases the functionally identical NALCN channels in the subthreshold range of membrane potential and consequently preserves normal pacemaking activity, demonstrating the flexible adaptive nature of the pacemaking in SNc DA neurons. This slow depolarization driven by multiple leak channels provides the reason why DA neuron pacemaking is robust and resistant to many kinds of pharmacological and molecular perturbations (*Guzman et al., 2009*). This is the first report to uncover molecular identities and properties of leak channels responsible for the pacemaking in SNc DA neurons.

SNc DA neurons express a variety of $K^+$, $Na^+$, $Ca^{2+}$, and NSCCs, and many ion channels and various factors contribute to pacemaking processes (*Gantz et al., 2018*). The voltage-dependent L-type $Ca^{2+}$ channels and HCN channels have long been the subject of controversy as the major pacemaker channels in SNc DA neurons (*Guzman et al., 2009*; *Surmeier et al., 2012*). The easiest and best way to find the ion channels responsible for pacemaking is to use specific blockers to see whether autonomous spontaneous firing is blocked. However, consistent with the latest report (*Guzman et al., 2009*), using both the brain slices and acutely dissociated DA neurons we came to the same conclusion that L-type $Ca^{2+}$ channels and HCN channels are not necessary for the pacemaking of SNc DA neurons. The HCN channels encoded by four genes (HCN1–4) are a hyperpolarization-activated nonselective cation channel and referred to as a pacemaker channel in many cells (*Zolles et al., 2006*; *Wahl-Schott and Biel, 2009*). They appear to have a wider range of activation and inactivation kinetics depending on the expression types, binding proteins, and many other factors such as several nucleotides and phosphatidylinositol-4,5-bisphosphate (PIP$_2$; *Zolles et al., 2006*; *Wahl-Schott and Biel, 2009*). Although HCN channels appear to have little participation in the pacemaking of the SNc DN neurons, they can be activated in response to a sudden and/or significant hyperpolarization of membrane potentials (*Gantz et al., 2018*). Therefore, HCN channels may play a role in stabilizing pacemaking activities upon sudden hyperpolarizing challenges. Regarding VACCs, Cav1.3 channels are known to be expressed abundantly in SNc DA neurons (*Gantz et al., 2018*). The Cav1.3 channel containing

α1D subunits can be characterized by low-voltage activation and slowly inactivating $Ca^{2+}$ currents, so it seems more ideal for slow depolarization of DA neurons than other members of the $Ca_v1$ family (*Xu and Lipscombe, 2001*). However, Cav1.3 channels give rise to intracellular $Ca^{2+}$ concentration ($[Ca^{2+}]_c$) via $Ca^{2+}$ influxes and form a microdomain with $Ca^{2+}$-activated SK channels in DA neurons (*Wolfart et al., 2001*). Therefore, SK channels have an opposite effect on $Ca^{2+}$ channel-mediated depolarization of the membrane potential (*Wolfart et al., 2001*). In addition, the continuous spontaneous firing of DA neurons itself causes continuous $Ca^{2+}$ influxes through many kinds of VACCs, thus the global $[Ca^{2+}]_c$ of DN neurons is always fluctuating and significantly higher than that in the TTX-treated silenced state (*Choi et al., 2003*). SK channels are abundantly expressed in SNc DA neurons (*Wolfart et al., 2001*) and can be already in a state of partial activation in such a $[Ca^{2+}]_c$ level of spontaneously firing DA neurons (*Choi et al., 2003*; *Kim et al., 2007*). It is, therefore, very likely that elevated $[Ca^{2+}]_c$ generates repolarizing outward $K^+$ currents, not only at microdomain levels but even at global levels. For this reason, $Na^+$ channels seem to be more important than $Ca^{2+}$ channels for the pacemaking of SNc DA neurons. In this aspect, the highly $Na^+$-permeable NALCN and TRPC3 channels appear to be ideal for driving pacemaking in SNc DN neurons.

In the midbrain DA neurons, insensitivity or resistance of the pacemaking to the antagonism of many ion channels have led to the multichannel hypothesis that the pacemaking activity results from the sum of activities of many ion channels (*Paladini et al., 2003*; *Chan et al., 2007*; *Guzman et al., 2009*). In addition, the slow depolarization in DA neurons appears to depend on a leak-like conductance (*Khaliq and Bean, 2010*). Therefore, many simulation models for DA neuron pacemaking have assumed a considerably large background leak conductance (*Guzman et al., 2009*; *Kuznetsova et al., 2010*). In fact, DA neurons have a very low input resistance ranging from 30 to 300 MΩ (*Grace and Bunney, 1983*; *Branch et al., 2014*), and large multipolar DA neurons receive several thousands of synaptic inputs throughout the somatodendritic compartment from network neurons (*Schultz, 1998*). Therefore, the DA neurons could be not only electrically very leaky but also geometrically unstable in terms of maintaining the homogenous membrane potential throughout the entire somatodendritic compartment (*Jang et al., 2014*). In such conditions, to generate and maintain a stable and steady rhythm of pacemaking, voltage-dependent or $Ca^{2+}$-permeable channels may not be a good choice. Therefore, unlike the classical pacemaker cells, the slow depolarization of DA neurons appears to depend on multiple voltage-independent $Na^+$-dependent leak channels, such as TRPC3 and NALCN channels. Although TRPC3 and NALCN channels belong to NSCCs, they primarily conduct $Na^+$ ions under physiological conditions (*Lu et al., 2007*; *Cochet-Bissuel et al., 2014*). The main characteristic of these two channels is their constitutive activation at very low membrane potentials and the linearity of the IV relationship in the subthreshold range of the membrane potential (*Dietrich et al., 2003*; *Zhou et al., 2008*; *Lu et al., 2007*). Nevertheless, because blocking one channel stops pacemaking completely, the pacemaking of SNc DA neurons depends on multiple channels in normal conditions. To analyze the relative contributions of TRPC3 and NALCN channels to the subthreshold depolarization, we measured the membrane potential shift from the $K^+$ equilibrium potential in the presence of TTX and ZD-7288. Given that blocking each of these channels leads to similar changes in the membrane potential shifts, TRPC3 and NALCN channels appear to equally contribute to slow depolarization in DA neurons. $Na^+$ substitution experiments show that these two channels are responsible for about 66.7 % of the subthreshold leak current, but do not cover the entire leak current completely. Still, some other channels are engaged in slow depolarization. Of course, HCN channels can also contribute to depolarization under excessively hyperpolarized conditions. In fact, when these two main channels are blocked completely, HCN channels can participate in the depolarization of membrane potential in the over-hyperpolarized conditions.

Although TRPC3 channel blockers completely stop autonomous firing in DA neurons, the pacemaking of DA neurons in TRPC3 KO mice is perfectly normal, indicating compensation by a functionally identical leak channel. In the neural intrinsic excitability, loss of specific ion channels can often be compensated by other channels due to homeostatic mechanisms (*Marder and Goaillard, 2006*). Since SNc DA neurons express a variety of ion channels including VACCs, HCN, and TRPC channels (*Chan et al., 2007*; *Branch et al., 2014*; *Tozzi et al., 2003*), we initially suspected that Cav1.3 channels, HCN channels, and TRPC channels may compensate the pacemaking in TRPC3 KO mice. However, all these channels do not appear to participate in the compensation of pacemaking in TRPC3 KO mice. In general, since TRPC3, TRPC6, and TRPC7 have similar primary sequences and activation

mechanisms and cluster closely on the phylogenic tree, TRPC6 and TRPC7 were prime candidates for compensation of the pacemaking in TRPC3 KO mice. However, although SKF96365 can block all members of TRPC channels, it did not have any effect on spontaneous firing in TRPC3 KO mice. Therefore, other members of the TRPC family do not appear to compensate the pacemaking in TRPC3 KO mice. Consistent with this result, RT-PCR experiments show that mRNAs of all TRPC channels are not increased in TRPC3 KO mice. On the other hand, the Na$^+$ leak channel NALCN is known to be widely expressed in the brain including DA neurons (*Ren, 2011*) and, therefore, has been suspected as a pacemaker channel in DA neurons (*Surmeier et al., 2012*; *Philippart and Khaliq, 2018*). Very recently, NBQN compounds were found to act as a NALCN channel blocker (*Hahn et al., 2020*). They do not affect TRPC channels but can partially inhibit voltage-dependent Na$^+$ and Ca$^{2+}$ channels in high concentrations. By taking this advantage, we find that NALCN is not only another important channel for pacemaking in DA neurons, but also compensates for TRPC3 leak currents in TRPC3 KO mice. NALCN and TRPC3 channels have many binding proteins and upstream regulators, including neurotransmitters and hormones (*Cochet-Bissuel et al., 2014*; *Amaral and Pozzo-Miller, 2007*). Therefore, the mechanism of pacemaking regulation can play diverse roles in DA neuron functions. Neurons in the diffuse modulatory systems, including DA neurons, receive and send information from and to wide areas of the brain, requiring a homogeneous and steady release of neurotransmitters to the extracellular fluid. Many of them are slow pacemaker neurons and need robust pacemaking. However, it is not yet clear whether TPRC3 and NALCN may play the same role in other slow pacemaker neurons, such as SNc DA neurons.

# Materials and methods

**Key resources table**

| Reagent type (species) or resource | Designation | Source or reference | Identifiers | Additional information |
|---|---|---|---|---|
| Antibody | Anti-TRPC3 (Rabbit polyclonal) | Alomone labs | Cat. #: ACC-016; RRID: AB_2040236 | IF (1:500) |
| Antibody | Anti-Tyrosine Hydroxylase (Mouse monoclonal) | Millipore | Cat. #: MAB318; RRID: AB_2201528 | IF (1:500) WB (1:1000) |
| Antibody | Anti-NALCN (Rabbit polyclonal) | Alomone labs | Cat. #: ASC-022; RRID: AB_11120881 | IF (1:1000) WB (1:1000) |
| Antibody | Anti-Mouse Alexa Fluor 488 (Goat polyclonal) | Thermo Fisher Scientific | Cat. #: A32723; RRID: AB_2633275 | IF (1:500) |
| Antibody | Anti-Rabbit Alexa Fluor 647 (Goat polyclonal) | Thermo Fisher Scientific | Cat. #: A32733; RRID: AB_2633282 | IF (1:500) |
| Antibody | Anti-Rabbit HRP (Goat polyclonal) | Bio-Rad | Cat. #: 170–6515; RRID: AB_11125142 | WB (1:1000) |
| Chemical compound, drug | Pyr3 | Tocris | Cat. #: 2004 | *Kiyonaka et al., 2009* |
| Chemical compound, drug | Pyr10 | Sigma-Aldrich | Cat. #: 648,494 | *Schleifer et al., 2012* |
| Chemical compound, drug | L-703,606 oxalate salt hydrate | Sigma-Aldrich | Cat. #: L119 | *Hahn et al., 2020* |
| Chemical compound, drug | Neurotensin | Sigma-Aldrich | Cat. #: N6383 | |
| Chemical compound, drug | Fluo-4, Pentapotassium Salt | Thermo Fisher Scientific | Cat. #: F14200 | |
| Chemical compound, drug | Oregon Green 488 BAPTA-1, Hexapotassium Salt | Thermo Fisher Scientific | Cat. #: O6806 | |
| Chemical compound, drug | Alexa Fluor 594 Hydrazide | Thermo Fisher Scientific | Cat. #: A10438 | |
| Chemical compound, drug | Streptavidin, Alexa Fluor 488 conjugate | Thermo Fisher Scientific | Cat. #: S11223 | |
| Chemical compound, drug | Neurobiotin | Vector laboratories | Cat. #: SP-1120 | |
| Chemical compound, drug | SR 95531 hydrobromide | Tocris | Cat. #: 1,262 | |

*Continued on next page*

*Continued*

| Reagent type (species) or resource | Designation | Source or reference | Identifiers | Additional information |
|---|---|---|---|---|
| Chemical compound, drug | CGP 55845 hydrochloride | Tocris | Cat. #: 1,248 | |
| Chemical compound, drug | NBQX disodium salt | Tocris | Cat. #: 1,044 | |
| Chemical compound, drug | (R)-CPP | Tocris | Cat. #: 0247 | |
| Chemical compound, drug | Isradipine | Tocris | Cat. #: 2004 | |
| Chemical compound, drug | ZD-7288 | Tocris | Cat. #: 1,000 | |
| Chemical compound, drug | Tetrodotoxin | Tocris | Cat. #: 1,078 | |
| Chemical compound, drug | 2-APB | Tocris | Cat. #: 1,224 | |
| Chemical compound, drug | SKF 96365 hydrochloride | Tocris | Cat. #: 1,147 | |
| Chemical compound, drug | Normal goat serum | Abcam | Cat. #: ab7481 | |
| Chemical compound, drug | Triton X-100 | Sigma-Aldrich | Cat. #: T8787 | |
| Chemical compound, drug | N-Methyl-D-glucamine | Glentham Life Sciences | Cat. #: GA0865 | |
| Strain, strain background (*M. musculus*) | Transgenic mouse line Th-EGFP, DJ76Gsat/Mmnc | MMRRC | PMID:26435058 | |
| Strain, strain background (*M. musculus*) | TRPC3 Knockout | PMID:18701065 | | LutzBirnbaumer (*Hartmann et al., 2008*) |
| Strain, strain background (*M. musculus*) | Crl:CD1(ICR) | Charles River Laboratories | Strain code: 022 | |
| Sequence-based reagent | GFP genotyping forward | This paper | PCR primers | CCT ACG GCG TGC AGT GCT TCA GC |
| Sequence-based reagent | GFP genotyping reverse | This paper | PCR primers | CGG CGA GCTGCA CGC TGC GTC CTC |
| Sequence-based reagent | TRPC3 genotyping forward | This paper | PCR primers | GAA TCC ACC TGC TTA CAA CCA TGT G |
| Sequence-based reagent | TRPC3 genotyping reverse | This paper | PCR primers | GGT GGA GGT AAC ACA CAG CTA AGC C- |
| Sequence-based reagent | Th forward | This paper | PCR primers | GCT GTG GCC TTT GAG AA |
| Sequence-based reagent | Th reverse | This paper | PCR primers | GCC AAG GAC AAG CTC AGG AA |
| Sequence-based reagent | TRPC1 forward | This paper | PCR primers | GCA AAC CCG TTT TGT TCG CA |
| Sequence-based reagent | TRPC1 reverse | This paper | PCR primers | AAA TGG AGT GGG CCA TGT GTA |
| Sequence-based reagent | TRPC2 forward | This paper | PCR primers | CTC AAG GGT ATG TTG AAG CAG T |
| Sequence-based reagent | TRPC2 reverse | This paper | PCR primers | AGC CGT CTT CCT GTT TGG TTC |
| Sequence-based reagent | TRPC3 forward | This paper | PCR primers | TGA CTT CCG TTG TGC TCA AAT ATG |
| Sequence-based reagent | TRPC3 reverse | This paper | PCR primers | CCT TCT GAA GCT TCT CCT TCT GC |
| Sequence-based reagent | TRPC4 forward | This paper | PCR primers | GCA AGA CAT TTC TAG CTT CCG C |
| Sequence-based reagent | TRPC4 reverse | This paper | PCR primers | GAG TAA TTT CTT CTT CGC TCT GGC |

*Continued on next page*

*Continued*

| Reagent type (species) or resource | Designation | Source or reference | Identifiers | Additional information |
|---|---|---|---|---|
| Sequence-based reagent | TRPC5 forward | This paper | PCR primers | TAC CAA TGT GAA GGC CCG AC |
| Sequence-based reagent | TRPC5 reverse | This paper | PCR primers | GCA TGA TCG GCA ATG AGC TG |
| Sequence-based reagent | TRPC6 forward | This paper | PCR primers | GCG CTC AGG TCA AGG TTC C |
| Sequence-based reagent | TRPC6 reverse | This paper | PCR primers | GTC ACC AAC TGA GCT GGA CC |
| Sequence-based reagent | TRPC7 forward | This paper | PCR primers | CTC CAA GTT CAG GAC TCG CT |
| Sequence-based reagent | TRPC7 reverse | This paper | PCR primers | GGG CCT TCA GCA CGT ATC TC |
| Sequence-based reagent | NALCN forward | This paper | PCR primers | CAA CAG CAA AAG GCA AGC GA |
| Sequence-based reagent | NALCN reverse | This paper | PCR primers | CCT ATG GCG GCT CAG TCA G |
| Sequence-based reagent | Th qRT-PCR forward | This paper | PCR primers | TGC TCT TCT CCT TGA GGG GT |
| Sequence-based reagent | Th qRT-PCR reverse | This paper | PCR primers | ACC TCG AAG CGC ACA AAG TA |
| Sequence-based reagent | GAPDH forward | This paper | PCR primers | GGA GAG TGT TTC CTC GTC CC |
| Sequence-based reagent | GAPDH reverse | This paper | PCR primers | ATG AAG GGG TCG TTG ATG GC-3 |
| Software, algorithm | Patchmaster | HEKA | RRID:SCR_000034 | |
| Software, algorithm | Fitmaster | HEKA | RRID:SCR_016233 | |
| Software, algorithm | LSM 510 meta | Zeiss | | |
| Software, algorithm | Origin 7.0 | Origin lab corporation | RRID:SCR_014212 | |
| Software, algorithm | IGOR Pro 4.01 | WaveMetrics | RRID:SCR_000325 | |
| Software, algorithm | QuantStudio 6 Real Time PCR System | Applied Biosystems | RRID:SCR_020239 | |
| Software, algorithm | Thermal Cycler Dice Real Time System III | TAKARA | | |
| Software, algorithm | Corel Graphics Suite 8 and 2019 | Corel Corporation | RRID:SCR_013674 | |
| Software, algorithm | Imaris | Bitplane | RRID:SCR_007370 | |
| Commercial assay or kit | HelixAmp Taq-plus with dye | Nanohelix | | Cat. #: PM001L |
| Commercial assay or kit | RNeasy Micro Kit | QIAGEN | | Cat. #: 74,004 |
| Commercial assay or kit | SYBR Green master mix | Applied Biosystems | | Cat. #: A25742 |
| Commercial assay or kit | TB Green premix Ex Taq | TAKARA | | Cat. #: RR420A |
| Commercial assay or kit | Superscript III for qRT-PCR | Thermo Fisher Scientific | | Cat. #: 11752050 |

## Animals

All experiments on animals were carried out in accordance with the approved animal care and use guidelines of the Laboratory Animal Research Center in Sungkyunkwan University School of Medicine (Suwon, Korea). Tyrosine hydroxylase-GFP (TH-GFP; CrljOri: CD1 background, DJ76Gsat/Mmnc MMRRC line, stock #1000292, inbred with Crl:CD1 mice) and TRPC3 knockout (129 sv/ev background, Birnbaumer Laboratories, NIH) (*Zhou et al., 2008*) mice were used regardless of sex. Mice were group

housed (two to five per cage, single sex) in a vivarium that was controlled for humidity, temperature (21°C–23 °C), and photoperiod (12 light/dark cycles). Mice had ad libitum access to food and water throughout the experiment. In all experiments, we used both male and female mice (postnatal days 18–26) weighing 9–12 g. Mice were identified by PCR of genomic DNA obtained by toe-biopsy. GFP or TRPC3 KO DNA were amplified using the following primers: GFP forward 5′-CCT ACG GCG TGC AGT GCT TCA GC-3′ and GFP reverse 5′ CGG CGA GCTGCA CGC TGC GTC CTC-3′ and TRPC3 KO forward: 5′-GAA TCC ACC TGC TTA CAA CCA TGT G-3′, and reverse: 5′-GGT GGA GGT AAC ACA CAG CTA AGC C-3′.

## Slice preparation

Horizontal midbrain slices were acutely prepared from male and female mice postnatal 18–26 days old. Mice were anesthetized with $CO_2$ gas and transcardially perfused with ice-cold high glucose artificial cerebrospinal fluid (aCSF in mM: 125 NaCl, 25 NaHCO$_3$, 2.5 KCl, 1.25 NaH$_2$PO$_4$, 0.4 Sodium ascorbate, 2 CaCl$_2$, 1 MgCl$_2$, and 25 D-glucose, pH 7.3 oxygenated with 95/5 % O$_2$/CO$_2$). After perfusion, brains were quickly removed and horizontally sliced in 250 µm thickness by using a VT-1000s vibratome (Leica, Germany), with high-glucose aCSF. After preparation of slices, slices were hemisected and superfused with normal aCSF (aCSF in mM: 125 NaCl, 25 NaHCO$_3$, 2.5 KCl, 1.25 NaH$_2$PO$_4$, 0.4 sodium ascorbate, 2 CaCl$_2$, 1 MgCl$_2$, and 10 D-glucose, pH 7.3 oxygenated with 95/5 % O$_2$/CO$_2$) at 30 °C. Recordings were performed from 40 min to 5 hr after removed from the bath.

## Acutely dissociated dopamine neuron preparation

To obtain dissociated dopamine neurons, whole brains were quickly removed from 18 to 26 days postnatal mice and immersed in ice-cold oxygenated (100 % O$_2$ gas) high-glucose HEPES-buffered saline, which contains (in mM: 135 NaCl, 5 KCl, 10 HEPES, 1 CaCl$_2$, 1 MgCl$_2$, and 25 D-glucose, pH adjusted to 7.4 with NaOH). Horizontal midbrain slices of 300 µm thickness were obtained with a TPI vibratome 1000 tissue sectioning system (TPI, USA). SNc regions from the slices were dissected out with a scalpel blade and digested with oxygenated high glucose HEPES-buffered saline containing papain (5–8 U/ml, Worthington, USA) for 20–30 min at 37 °C. Next, the slice segments were rinsed with enzyme-free HEPES-saline, and then tissues were gently agitated with varying sizes of fire-polished Pasteur pipettes. The agitated cells were gently attached to a poly-D-lysine (0.01%)-coated glass coverslip for 30 min at room temperature. Recordings were performed from 40 min to 3 hr after being attached.

## Electrophysiological recording

For brain slice recording, slice tissues were transferred to a submerged slice chamber and continuously perfused with warm (33 °C) oxygenated (95/5 % O$_2$/CO$_2$ gas) aCSF. To exclude synaptic inputs in slice recordings, glutamate and GABA receptor blockers (10 µM NBQX, Tocris, UK, 1044; 5 µM CPP, Tocris, 0247; 5 µM SR-95531, Tocris, 1262; 1 µM CGP-55845, Tocris, 1248) were included in the recording solutions.

Dopamine neurons were identified by GFP fluorescence using the HBO 100 microscope illuminating system (Zeiss, Germany) and a UM-300 CCD camera (UNIQ Vision, USA) on the Axioskop two microscope equipped with a W plan-apochromat 20× lens (Zeiss). To ensure the drug delivery for target neurons, cell bodies were selected near 50–100 µm from the surface of tissues.

Single dissociated dopamine neurons were perfused with normal HEPES-buffered saline containing (in mM 140 NaCl, 5 KCl, 10 HEPES, 1 CaCl$_2$, 1 MgCl$_2$, and 10 D-glucose, pH adjusted to 7.4 with NaOH at room temperature [~23 °C]). Neurons were visualized using an IX70 Olympus microscope equipped with a Photometrics Quantix CCD camera (Roper Scientific Inc, USA). To select GFP-positive dopamine neurons, dissociated neurons were exposed to 480 nm light generated by a High-Power LED collimator source (Mightex, USA) with a Universal LED controller (Mightex).

Low-resistance patch electrodes (3–4 MΩ) were pulled from filamented borosilicate patch electrodes (World Precision Instruments, USA) with a PC-100 micropipette puller (Narishige, Japan). Electrophysiological signals were recorded with an EPC-10 or EPC-9 amplifier (HEKA Elektronik, Germany), low-passed filtered at 1–2 kHz and digitized at 10–20 kHz with the Patch Master software (HEKA Elektronik). Whole-cell patch recordings were performed on a conventional tight giga-ohm seal (>3 GΩ), and the patch electrode was filled with normal potassium-based internal solutions (in mM: 120 K-gluconate, 20 KCl, 10 HEPES, 2 Mg-ATP, 0.3 Na-GTP, 4 Na-ascorbate, 10 Na$_2$-phosphocreatine,

pH adjusted to 7.3 with KOH and osmolarity about 290 mOsm/Kg). Seal resistance was monitored by a 100 ms test pulse of 10 mV and reported voltages were corrected for a liquid junction potential of ~−15 to ~−20 mV between internal and external solution in which measured using a flowing 3 M KCl electrode. For firing recording in single dissociated neurons, loose-seal cell-attached recordings ($R < 100$ MΩ) were performed in a current clamp configuration.

External sodium removal experiments were performed by NMDG-based external solutions contained in mM: 151 NMDG, 3.5 KCl, 2 CaCl$_2$, 1 MgCl$_2$, 10 glucose, and 10 HEPES, pH adjusted to pH 7.4 with HCl. The membrane potential in NMDG-based external solutions were estimated by the following Goldman-Hodgkin-Katz equation as follows: $V_m = RT/F \ln(p_K[K^+]_o + p_{Na}[Na^+]_o + p_{Cl}[Cl^-]_i / p_K[K^+]_i + p_{Na}[Na^+]_i + p_{Cl}[Cl^-]_o)$. $p_K$, $p_{Na}$, and $p_{Cl}$ are relative membrane permeability for $K^+$, $Na^+$, and $Cl^-$. The $p_K$, $p_{Na}$, and $p_{Cl}$ are relative membrane permeability for $K^+$, $Na^+$, and $Cl^-$, and the ratio of $p_K : p_{Na} : p_{Cl}$ used was 1:0.05:0.45 (*Goldman, 1943*; *Schleifer et al., 2012*).

For the application of agonists by pressure microinjection 'puff experiments', we used a IM 300 microinjector (Narishige). The injection glass pipette filled with the HEPES-buffered saline containing 10 μM neurotensin (Sigma-Aldrich, N6383) was positioned near proximal dendrites about >20 μm of target neurons to prevent the mechanical effect. The single-pulse duration was 1 s, during which time the pressure was 200–300 kPa.

## Two-photon microscopy imaging

For fluorescence imaging, dopaminergic neurons were loaded with Alexa Flour 594 (Alexa-594, 30 μM, Thermo Fisher, USA, A10438) and Oregon Green BAPTA-1 (OGB-1, 200 μM, Thermo Fisher, O6806) or Fluo-4 (200 μM, Thermo Fisher, F14200) via patch electrode. The two-photon excitation source was a Ti::sapphire laser system (Mai Tai, 690–1020 nm, Spectra-physics, USA). Optical signals were acquired using an 800 nm excitation beam that simultaneously excites Alexa-594 and OGB-1 (or Fluo-4) dyes. The laser-scanned images were acquired with an LSM510 Meta system (Zeiss). Ca$^{2+}$ imaging was allowed 10 min after the whole-cell break-in for dye equilibration. Frame scan images were acquired at 10–20 ms per frame at 512 pixels dwell time. Dendritic Ca$^{2+}$ changes were presented as a ratio ($\Delta G/\Delta R = (G-G_{min})/(R-R_{min})$) after background-subtraction, in which calcium indicator fluorescence ($\Delta G$) was divided by Alexa-594 ($\Delta R$).

## Fixed tissue preparation and imaging

Brain slices were fixed with 4 % paraformaldehyde (PFA) in ice-cold 1× phosphate-buffered saline (PBS, in mM: 137 NaCl, 2.7 KCl, 10 Na$_2$HPO$_4$, KH$_2$PO$_4$, pH adjusted to 7.4 with NaOH). Fixed slices were washed with PBS and incubated with 2 % normal goat serum (NGS) and 0.1 % Triton X-100 in PBS for 30 min at room temperature. After rinsing, tissue slices were incubated with primary antibody in 2 % NGS in PBS overnight. The primary antibodies were mouse-anti-TH (1:500, Millipore, USA, MAB318, RRID: AB_2201528) and rabbit-anti-TRPC3 (1:500, Alomone lab, Israel, ACC-016, RRID: AB_2040236). After the primary antibody reaction, tissues were washed three times with NGS in PBS solutions at room temperature for 30 min. Next, the secondary antibody was treated to slices at room temperature for 2 hours. The secondary antibodies were anti-Alexa- rabbit-647 (1:500, Thermo Fisher, USA, A32733; RRID: AB_2633282) or anti-mouse-448 (1:500, Thermo Fisher, USA, A32723; RRID: AB_2633275) or streptavidin-594 (1:2000, Thermo Fisher, USA, S11227). After secondary antibody reaction, tissues were rinsed again with 2 % NGS solution. Finally, samples were acquired using Mai Tai two-photon laser using 760 nm of excitation beam on the Axioskop 2 microscope by 20× W plan-apochromat lens (Zeiss, Germany). The frame-scanned images were acquired on the LSM 510 Meta as Z-section at 0.5 μm × 50–100 sections, and each image was obtained by averaging four images acquired with a resolution of 1024 × 1024 pixels. Z-section images were reconstructed to 3D by using IMARIS 7.0 (Bitplane, USA).

## Single-cell RT-PCR and qRT-PCR

Th-eGFP-positive single dissociated neurons were aspirated into a microelectrode pipette with a sampling solution containing 10 mM dithiothreitol (DTT), 50 U/ml RNasin RNase inhibitor (Promega, USA, N2611) in diethylene pyrocarbonate-treated water. After collecting sample cells, RNA was extracted using RNEasy kits (Qiagen, Germany, 744004). For single-cell RT or qRT-PCR, RNA was purified by ethanol precipitation procedures (*Liss, 2002*). Purified RNA was heated to 50 °C for 30 min.

cDNA was synthesized from cellular mRNA through the addition of SuperScript III Kits (Thermo Fisher, 11752050). The reaction mixture was incubated sequentially at 25 °C for 10 min, 50 °C for 30 min, and then heated to 85 °C for 5 min. After reverse transcription, cDNA samples were chilled at 4 °C. For single-cell qPCR, cDNA was purified according to a published procedure (*Liss, 2002*). After purification, a single-cell cDNA sample was used as a template for conventional PCR amplification. Cycle conditions were as follows: 95 °C for 15 s, 55 °C for 30 s, 72 °C 45 s. RT-PCR was performed using the following primers (forward followed by reverse): Th: 5′-GCT GTG GCC TTT GAG AA-3′ and 5′-GCC AAG GAC AAG CTC AGG AA-3′; trpc3: 5′-TGA CTT CCG TTG TGC TCA AAT ATG-3′ and 5′-CCT TCT GAA GCT TCT CCT TCT GC-3′. The qRT-PCR was conducted with cDNA and TB Green premix (Takara, Japan, RR420A) or SYBR Green master mix (Applied Biosystems, USA, A25742) and 50 cycles of 95 °C for 15 s, 60 °C for 20 s, 72 °C for 15 s performed using a Takara Thermal Dice Real-time system III (Takara) or a QuantStudio 6 Flex Real-time PCR system (Applied Biosystems, USA). For qRT-PCR, the following primers were used (forward followed by reverse): NALCN: 5′-CAA CAG CAA AAG GCA AGC GA-3′ and 5′-CCT ATG GCG GCT CAG TCA G-3′; trpc1: 5′-GCA AAC CCG TTT TGT TCG CA-3′ and 5′-AAA TGG AGT GGG CCA TGT GTA-3′; trpc2: 5′-CTC AAG GGT ATG TTG AAG CAG T-3′ and 5′-AGC CGT CTT CCT GTT TGG TTC-3′; trpc4: 5′-GCA AGA CAT TTC TAG CTT CCG C-3′ and 5′-GAG TAA TTT CTT CTT CGC TCT GGC-3′; trpc5: 5′-TAC CAA TGT GAA GGC CCG AC-3′ and 5′-GCA TGA TCG GCA ATG AGC TG-3′; trpc6: 5′-GCG CTC AGG TCA AGG TTC C-3′ and 5′-GTC ACC AAC TGA GCT GGA CC-3′; trpc7: 5′-CTC CAA GTT CAG GAC TCG CT-3′ and 5′- GGG CCT TCA GCA CGT ATC TC-3′; TH: 5′-TGC TCT TCT CCT TGA GGG GT-3′ and 5′-ACC TCG AAG CGC ACA AAG TA-3′; GAPDH: 5′-GGA GAG TGT TTC CTC GTC CC-3′ and 5′-ATG AAG GGG TCG TTG ATG GC-3′. The analysis was performed according to the $\Delta C_t$ and $\Delta\Delta C_t$ method. Cycle threshold ($C_t$) values of test genes were subtracted by GAPDH or tyrosine hydroxylase reference genes ($\Delta C_{t\,sample} = C_{t\,target} - C_{t\,ref}$). To compare relative gene expression in tissue samples, each $\Delta C_t$ values of samples normalized to the NALCN $\Delta C_t$ values ($\Delta\Delta C_t$).

## Western blotting

For western blotting, SNc tissue slices were collected from the littermates of WT TRPC3 KO hetero (+/−) and homo (−/−) mice and lysed on ice for 30 min in protein lysis buffer (1 mM $Na_3VO_4$, 1 mM NaF, complete protease inhibitor cocktail (Roche, USA), 1 % Triton X-100 in PBS). The protein samples were separated by SDS–PAGE and transferred to a hydrophobic polyvinylidene difluoride (PVDF) membrane. The membranes were blocked in 5 % skim milk in Tween tris-buffered saline (TTBS) for 1 hr at room temperature and then incubated in anti-NALCN (1:1000, Alomone Labs, Israel, ASC-022; RRID: AB_11120881) and anti-TH (1:1000, Millipore, USA, MAB318; RRID: AB_2201528) for 18 hr at 4 °C. After three washes, the membranes were incubated for 2 hr at room temperature with HRP-conjugated rabbit secondary antibody (Bio-Rad, USA, 170–6515; RRID: AB_11125142) for NALCN detection and HRP-conjugated mouse secondary antibody (Bio-Rad, 170–6516; RRID: AB_11125547) for pan Cadherin detection, and then washed. The blot was developed using enhanced chemiluminescence reagent (GE Healthcare, USA).

## Pharmacological reagents

All chemicals were purchased from Sigma (USA), except tetrodotoxin (Tocris, Cat. #: 1078), isradipine (Tocris, Cat. #: 2004), pyr3 (Tocris, Cat. #: 2004), ZD-7288 (Tocris, Cat. #: 1000), SKF 96365 (Tocris, Cat. #: 1147), 2-APB (Tocris, Cat. #: 1224), CPP (Tocris, Cat. #: 0247), NBQX (Tocris, Cat. #: 1044), SR 95531 (Tocris, Cat. #: 1262), and CGP-55845 (Tocris, Cat. #: 1248), pyr10 (Millipore, USA, Cat. #: 648494), Fluo-4, OGB-1, and Alexa-594 (Thermo Fisher, USA). Drug stock solutions were prepared in DMSO or deionized water, and stocks were diluted to final concentrations in external solutions via sonication for 10 min.

## Quantification and statistical analysis

All graphical illustrations were performed using CorelDraw 8 and 2019 software (Corel Corporation, USA). For drug applications, 20 s in each condition (control or drug) was used for measuring the mean of firing frequency. All data collected were analyzed using Origin 7.0 software (Origin Lab Corporation, USA), and electrophysiological data were analyzed using Igor Pro 4.01 (Wavemetrics, USA). All numeric data are presented as mean ± standard error of the mean (S.E.M). Data were summarized

as box plots, with the centerline showing the median, the top and bottom of the box indicating the 25–75% range, and whisker representing the 5–95% range. For comparison of data, one-way analysis of variance (ANOVA) was performed to assess the statistical significance of the difference between groups and p-values were significant at *$p<0.05$; **$p<0.01$; ***$p<0.001$.

## Acknowledgements

Funding for this research was supported by the National Research Foundation of Korea (NRF) grant funded by the Korea government (MSIT) (No. 2017R1A2B3005656). LB was supported in part by the Intramural Research Program of the NIH (project Z01ES101648).

## Additional information

### Funding

| Funder | Grant reference number | Author |
|---|---|---|
| National Research Foundation of Korea | 2017R1A2B3005656 | Myoung Kyu Park |
| National Institutes of Health | Z01ES101648 | Lutz Birnbaumer |

The funders had no role in study design, data collection and interpretation, or the decision to submit the work for publication.

### Author contributions

Ki Bum Um, Conceptualization, Data curation, Formal analysis, Methodology, Software, Validation, Visualization, Writing - original draft, Writing - review and editing; Suyun Hahn, Data curation, Methodology, Validation; So Woon Kim, Data curation, Formal analysis, Methodology; Yoon Je Lee, Data curation, Formal analysis; Lutz Birnbaumer, Resources, Writing - review and editing; Hyun Jin Kim, Conceptualization, Validation, Writing - review and editing; Myoung Kyu Park, Conceptualization, Funding acquisition, Methodology, Project administration, Validation, Writing - original draft, Writing - review and editing

### Author ORCIDs

Ki Bum Um http://orcid.org/0000-0002-4882-5736
Hyun Jin Kim http://orcid.org/0000-0002-5806-4416
Myoung Kyu Park http://orcid.org/0000-0001-8111-6096

### Ethics

This study was performed in strict accordance with the recommendations in the Guide for the Care and Use of Laboratory Animals Research Center (LARC) in Sungkyunkwan University. All of the animals were handled according to approved institutional animal care and use committee (IACUC) protocols (SKKU IACUC2021-03-11-1) of Sungkyunkwan University.

### Decision letter and Author response

Decision letter https://doi.org/10.7554/eLife.70920.sa1
Author response https://doi.org/10.7554/eLife.70920.sa2

## Additional files

### Supplementary files

• Transparent reporting form

### Data availability

Data used to generate summary plots presented in Figures 1-6 are included in the manuscript and are provided as source data files.

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
