## [Decision Letter]

**Acceptance summary:**

This work clearly demonstrates an important role for two specific sodium-permeable ion channels for maintaining the pacemaker-like firing of midbrain dopamine neurons. These neurons have a key role in motivation, reinforcement and locomotion, and have been implicated in Parkinson's disease and multiple neuropsychiatric disorders. The authors also find that the regular firing of these cells is robustly maintained even when one of the two channels is knocked out, through upregulation of the level of the other channel. The authors have added new data and discussion to address concerns of the reviewers.

**Decision letter after peer review:**

Thank you for submitting your article "TRPC3 and NALCN channels drive pacemaking in substantia nigra dopaminergic neurons" for consideration by *eLife*. Your article has been reviewed by 3 peer reviewers, one of whom is a member of our Board of Reviewing Editors, and the evaluation has been overseen by Kenton Swartz as the Senior Editor. The following individual involved in review of your submission has agreed to reveal their identity: Gary Yellen (Reviewer #3).

The reviewers have discussed their reviews with one another, and the Reviewing Editor has drafted this to help you prepare a revised submission. The reviewers felt that this manuscript does an excellent job of clarifying the roles of TRPC3 and NALCN channels in the pacemaking activity of midbrain dopamine neurons.

Essential revisions:

1) There is one point that would be improved with a further experiment. Many experiments in the paper use the 0 Na (NMDG) condition to estimate the maximal effect of Na-conducting channels on the membrane potential; these are not crucial to the main conclusions, but they are a basis for the quantitative estimates of the contributions of different currents. An important concern is that under these conditions, the Na-Ca exchanger will cease to function or will function in the reverse direction from the normal outward pumping of Ca. Particularly in high Ca load cells like the SNc DA neurons, this can lead to very large increases in intracellular Ca. This could alter the membrane potential, particularly by activating SK channels that will hyperpolarize the neurons. This is a possible confound for measuring the contributions of the different inward currents. To learn whether this is a problem, the authors should apply SK blockers to see if the NMDG hyperpolarization is smaller. It might also be advisable to perform Ca imaging (as in Figure 1) to see if the NMDG condition elevates intracellular Ca.

Other suggestions can be found in the individual reviewer comments below, but are not felt to be critical to support the main conclusions of the paper (Recommendations for the authors).

*Reviewer #1 (Recommendations for the authors):*

1. Figure 2 illustrates that there are multiple ion channels that can contribute to pacemaker activity in the dopamine neurons. Figure 2 shows that with TRPC3 channels blocked pharmacologically, injection of DC current is sufficient to return the neuron to pacemaking spiking with identical time course (at least in the illustrated cell). The same observation (Figure 4) is seen with NALCN channels blocked. Is the identical timecourse of the action potential always observed? A figure quantifying this would emphasize this point, showing the time course of the depolarizing phases and the depth of the AHP for a number of cells.

2. To explain the data in Figure 6A, showing that addition of the HCN blocker ZD has a strong hyperpolarizing effect when added after blockade of TRPC3 and NALCN channels, the authors suggest that the membrane is now in a range where HCN channels are activated. However, in the panels next to this with either pyr-10 or L-703 alone, the membrane potential is the same as with both blockers, yet ZD has no further effect. The neuron is clearly in the same voltage range – why would ZD have this effect only in the presence of both other blockers?

3. Can the authors rule out that autocrine activation of D2 receptors is not involved in the apparent effects of these drugs, given that Phillipart and Khaliq demonstrated a strong effect of D2R activation on NALCN currents?

4. The rationale for using brain slices vs. dissociated cells in particular experiments should be provided. It would be especially nice to see the data in Figure 4E-G in slices from WT vs. TRPC3 KO mice.

*Reviewer #2 (Recommendations for the authors):*

1) I suggested providing a more nuanced description of the inhibitors used, in particular Py10, because the evidence for its specificity for TRPC3 is not strong.

2) The authors should indicate how many mice were used in the recordings in all figure panels.

3) The authors should make sure that it is clearly indicated in each figure legend which data were obtained from slices and which from acutely dissociated neurons.

4) The authors should explicitly indicate whether the voltage traces with and without inhibitor (e.g. Figure 1E) were obtained from the same neuron.

5) I think it could be interesting for readers if the authors provided a brief discussion on the presence of TRPC3 and NALCN channels in human dopaminergic neurons.

*Reviewer #3 (Recommendations for the authors):*

There are many experiments here that use the 0 Na (NMDG) condition to estimate the maximal effect of Na-conducting channels on the membrane potential; these are not crucial to the main conclusions, but they are a basis for the quantitative estimates of the contributions of different currents. An important concern about these experiments is that under these conditions, the Na-Ca exchanger will cease to function or will function in the reverse direction from the normal outward pumping of Ca. Particularly in high Ca load cells like the SNc DA neurons, this can lead to very large increases in intracellular Ca. This could alter the membrane potential, particularly by activating SK channels that will hyperpolarize the neurons.

This is a possible confound for measuring the contributions of the different inward currents. To learn whether this is a problem, the authors should apply SK blockers to see if the NMDG hyperpolarization is smaller. It might also be advisable to perform Ca imaging (as in Figure 1) to see if the NMDG condition elevates intracellular Ca.

Other comments:

Figure 1C: Presumably the pie chart refers to the durations of the three phases; the legend should say this.

Figure 1D: Why is the distribution of firing rates here so different from the typical firing rates in (e.g.) Figure 2B?

Figure 1G: How are these data normalized (i.e. what is the x-axis)?

Figure 1H: The immunostaining for the TRPC3 channel is no more membrane-localized than that for the cytosolic TH enzyme, which seems strange unless most of the channels are in an intracellular pool. The specificity of the antibody should be checked by staining the TRPC3 knockout.

Figure 1J: the legend says that the figure shows RT-PCR for both TRPC3 and TH, but only one dataset is shown.

---

## [Author Response]

Essential revisions:1) There is one point that would be improved with a further experiment. Many experiments in the paper use the 0 Na (NMDG) condition to estimate the maximal effect of Na-conducting channels on the membrane potential; these are not crucial to the main conclusions, but they are a basis for the quantitative estimates of the contributions of different currents. An important concern is that under these conditions, the Na-Ca exchanger will cease to function or will function in the reverse direction from the normal outward pumping of Ca. Particularly in high Ca load cells like the SNc DA neurons, this can lead to very large increases in intracellular Ca. This could alter the membrane potential, particularly by activating SK channels that will hyperpolarize the neurons. This is a possible confound for measuring the contributions of the different inward currents. To learn whether this is a problem, the authors should apply SK blockers to see if the NMDG hyperpolarization is smaller. It might also be advisable to perform Ca imaging (as in Figure 1) to see if the NMDG condition elevates intracellular Ca.Other suggestions can be found in the individual reviewer comments below, but are not felt to be critical to support the main conclusions of the paper (Recommendations for the authors).

To address this issue, we have performed additional experiments as the reviewer suggested (Figure 3—figure supplement 4). To examine whether the membrane potential in the extracellular 0 mM Na^+^ condition (NMDG substitution) might be affected by SK channels, we applied the SK channel blocker apamin (100 nM) after the complete replacement of Na^+^ with NMDG in the presence of TTX (500 nM). As shown in Figure 3—figure supplement 4, application of apamin did not affect the membrane potential at all (NMDG-induced hyperpolarization = -71.81 ± 1.53 mV, NMDG and apamin-induced hyperpolarization = -72.50 ± 1.53 mV; p = 0.711, *n* = 7 from 3 mice). In addition, when we measured intracellular ca^2+^ concentration ([ca^2+^]_c_) simultaneously, Na^+^ replacement with NMDG decreased [ca^2+^]_c_ rather than increased [ca^2+^]_c_. These results suggest that Na^+^/ca^2+^ exchangers in the 0 mM Na^+^ condition have little effect on the membrane potentials that we measured. Because Na^+^ replacement with NMDG hyperpolarized the membrane potential, it could close many kinds of voltage-activated ca^2+^ channels (mainly L-type ca^2+^ channels in DA neurons), consequentially decreasing both ca^2+^ influxes and [ca^2+^]_c_. We added these results as a new Figure 3—figure supplement 4 and added a brief explanation in the manuscript as follows:

Results section: “When all extracellular Na^+^ was replaced by equimolar NMDG after treatment with pyr10, the membrane potential of DA neurons was maximally hyperpolarized (Figures 3E and 3F). In the absence of extracellular Na^+^, Na^+^/ca^2+^ exchangers may activate SK channels by increasing intracellular ca^2+^ concentration ([ca^2+^]_c_) and then affect the membrane potential that we measured. Therefore, we examined whether the SK channel blocker apamin (100 nM) affects the membrane potential when extracellular Na^+^ was replaced with NMDG. However, there was no change in the membrane potentials, suggesting that this was not the case (Figure 3—figure supplement 4). These data suggest that the sustained inward currents produced by TRPC3 channels must be compensated by other Na^+^-permeable ion channels in TRPC3 KO mice.”

Reviewer #1 (Recommendations for the authors):1. Figure 2 illustrates that there are multiple ion channels that can contribute to pacemaker activity in the dopamine neurons. Figure 2 shows that with TRPC3 channels blocked pharmacologically, injection of DC current is sufficient to return the neuron to pacemaking spiking with identical time course (at least in the illustrated cell). The same observation (Figure 4) is seen with NALCN channels blocked. Is the identical timecourse of the action potential always observed? A figure quantifying this would emphasize this point, showing the time course of the depolarizing phases and the depth of the AHP for a number of cells.

We have analyzed time courses of the firings regenerated by DC current injection under the inhibition of TRPC3 or NALCN in more detail (TRPC3, Figure2—figure supplement 1; NALCN, Figure 4—figure supplement 2). Since the slow depolarization during the interspike interval consists of three phases, we measured each duration. In addition, we also compared slopes of the phase II depolarization and AHP values in phase I. However, when the regenerated firing rate was adjusted to the same control level, we cannot find any significant difference between the control and the regenerated action potentials. Therefore, we added two supplementary figures (Figure 2-supplement 1 and Figure 4-supplement 2) in the manuscript:

Results section: “When the firing rate was resuscitated to the control level, voltage traces of the regenerated slow depolarization and action potential were completely aligned with those before pyr10 treatment (Figure 2A, right bottom; Figure 2-supplement 1), indicating that the channel inhibited by pyr10 could be a leak-like channel. Consistent with these data, the firing rate was gradually increased within the pacemaking range by a slow ramp-like increase in current injection (Figure 2A, right-top), implying that the amount of leak current determines the pacemaking rate in SNc DA neurons.”

Results section: “When the firing rate was resuscitated to the control level, the voltage traces between the revived firings and those before L-703,606 treatment were completely aligned to each other (Figure 4C, Figure 4-supplement 2), indicating that the pacemaking can be completely revived by a leak-like current.”

2. To explain the data in Figure 6A, showing that addition of the HCN blocker ZD has a strong hyperpolarizing effect when added after blockade of TRPC3 and NALCN channels, the authors suggest that the membrane is now in a range where HCN channels are activated. However, in the panels next to this with either pyr-10 or L-703 alone, the membrane potential is the same as with both blockers, yet ZD has no further effect. The neuron is clearly in the same voltage range – why would ZD have this effect only in the presence of both other blockers?

HCN channels in SNc DA neurons appear to depolarize the membrane potential in a significantly hyperpolarized condition. Midbrain DA neurons express HCN2-HCN4 subunits and they begin to activate below -50 mV with a half-maximal activation at –65 mV (Zolles et al., Neuron, 2006, 52 (6):1027-1036). In Figure 1—figure supplement 1F, we show that HCN channels are partially and transiently activated in the lowest part of the AHP below -65 mV in the phase I of the interspike interval. However, activation of HCN channels during the interspike interval is so weak that they cannot change normal pacemaking rate of SNc DA neurons. In Figure 6, since we treated TTX and measured the steady-state membrane potential, the action of HCN channels for the membrane potential may differ from its natural dynamic condition. However, it was possible to directly compare TRPC3 and NALCN currents using specific blockers in this particular condition. They appear to act like ‘leak currents’ and have a linear I-V relationship within the subthreshold membrane potential range (thus additive each other as a depolarizing current). Since pyr-10 or L703,606 similarly hyperpolarized the membrane potential, TRPC3 and NALCN currents contribute equally to the depolarization in the subthreshold range of the membrane potential. Blocking only one of them hyperpolarized the membrane potential up to -60 mV, which is insufficient to significantly activate HCN channels (Figure 6A, black traces). However, when both of these channels are inhibited, the membrane potential is hyperpolarized to more than -70 mV, sufficiently activating HCN channels (Figure 6A, red traces). Therefore, ZD-7288 can affect the membrane potential when both other blockers are present.

3. Can the authors rule out that autocrine activation of D2 receptors is not involved in the apparent effects of these drugs, given that Phillipart and Khaliq demonstrated a strong effect of D2R activation on NALCN currents?

In one of our previous studies (Jang et al., 2011, J Neurochem. 116:966-974,), we reported the functional role of D2 autoreceptors on the regulation of spontaneous firings in acutely dissociated SNc DA neurons. Using the cell-attached patch-clamp technique, application of the D2 antagonist sulpiride did not affect pacemaking in acutely dissociated SNc DA neurons, suggesting that there is no possible autocrine effect of D2 receptors on the pacemaking activity in acutely dissociated DA neurons. Therefore, in this study, many critical experiments about effects of drugs and chemicals had been performed not only in the midbrain slices, but also in acutely dissociated DA neurons (Figure 1-supplement 2, Figure 3-supplement 2, Figure 4-supplement 1).

4. The rationale for using brain slices vs. dissociated cells in particular experiments should be provided. It would be especially nice to see the data in Figure 4E-G in slices from WT vs. TRPC3 KO mice.

Brain slices have long been used as a standard technique by many electrophysiologists in neuroscience to study a variety of neuronal properties. A major advantage of brain slices compared to acutely dissociated neurons is that experiments can be performed in better physiological conditions due to the maintenance of the intrinsic structural and functional integrity of synaptic connections. Therefore, electrophysiological recordings in brain slices are generally more stable than those in acutely dissociated neurons. However, most neurons reside deep in the slice tissues, making it difficult to achieve precise concentrations of some drugs and chemicals. In particular, many ion channel blockers have to diffuse tens of micrometers into the slice and bind specific sites buried in the lipid bilayer. Therefore, it takes a much longer time and high concentrations of drugs have been often used. In addition, target neurons in the slices may also be connected to nearby active network neurons and receive constant inputs, and perhaps exposed to ambient neurotransmitters and neurohormones. In this aspect, the acutely isolated neurons are free from diffusion problems of drugs and environmental factors including synaptic inputs, neurohormones, and neurotransmitters. It also allows precise control of the concentration of drugs and chemicals on target ion channels. Especially, it is very useful when investigating intrinsic electrical properties (i.e. pacemaking) which depend on the activities of intrinsic ion channels in the neuronal cell membrane. However, the main disadvantage of using acutely dissociated neurons is the difficulty in isolating neurons and the sufficient number of neurons required. Therefore, by performing channel antagonists experiments in both conditions, we were more confidently and clearly able to find two ion channels essential for pacemaking of SNc DA neurons.

Reviewer #2 (Recommendations for the authors):1) I suggested providing a more nuanced description of the inhibitors used, in particular Py10, because the evidence for its specificity for TRPC3 is not strong.

We changed some sentences in the manuscript as follows:

“Therefore, we used pyrazole derivatives such as pyr3 and pyr10 to selectively inhibit TRPC3 channels, although pyr10 is reported to partially inhibit TRPC6, too (Kyonaka et al., 2009; Schleifer et al., 2012). These channel blockers completely abolished spontaneous firing of DA neurons in the midbrain slices (Figures 1E and 1F), together with the disappearance of dendritic ca^2+^ oscillations, which were measured by Fluo-4 in the whole-cell patch pipette (Figures 1E and 1G).”

2) The authors should indicate how many mice were used in the recordings in all figure panels.

In all experiments, we used more than 3 mice. We have added the correct mice number in each figure legend.

3) The authors should make sure that it is clearly indicated in each figure legend which data were obtained from slices and which from acutely dissociated neurons.

We indicated in each figure legend.

4) The authors should explicitly indicate whether the voltage traces with and without inhibitor (e.g. Figure 1E) were obtained from the same neuron.

We added these descriptions to the figure legend.

5) I think it could be interesting for readers if the authors provided a brief discussion on the presence of TRPC3 and NALCN channels in human dopaminergic neurons.

Since TRPC3 and NALCN are evolutionarily conserved in structure (Tang et al., 2018, Cell Res., 2018 Jul;28(7):746-755; Kschonsak et al., 2020, Nature, Nov;587(7833):313-318), we believe that these channels could play similar roles in human DA neurons. There is also increasing evidence for many pathophysiological roles of TRPC3 (Zhou et al., 2008, J. Neuroscience, Jan 9;28(2):473-82; Ju et al., 2015, Front Physiol. Mar 25;6:86; Qi et al., 2016, Int. J. Cardiol., 2019, Jan 15;203:169-81; Hof et al., Nature Reviews, 2019, Jun;16(6):344-360) and NALCN (Lu et al., Mol Neurobiol., 2012, Jun;45(3):415-23; Cochet-Bissuel et al., Front Cell Neurosci., 2014, May 20;8:132; Lutas et al., *eLife*, 2016, May 13;5:e15271; Chua et al., Science Advances, 2020, May 13;5:e15271) channels in several autonomously active neurons. TRPC3 (Cederholm et al., The cerebellum, 2019, Jun;18(3):536-543; proteinatlas.org, ENSG00000138741-TRPC3) and NALCN are expressed in the human midbrain (proteinatlas.org, ENSG00000102452-NALCN) (Thul & Lindskog, Protein Sci. 2018 Jan; 27(1): 233–244). However, little is known about TRPC3 and NALCN in human DA neurons yet. Nevertheless, we expect that the importance of TRPC3 and NALCN channels in human DA neurons will emerge in the near future.

Reviewer #3 (Recommendations for the authors):There are many experiments here that use the 0 Na (NMDG) condition to estimate the maximal effect of Na-conducting channels on the membrane potential; these are not crucial to the main conclusions, but they are a basis for the quantitative estimates of the contributions of different currents. An important concern about these experiments is that under these conditions, the Na-Ca exchanger will cease to function or will function in the reverse direction from the normal outward pumping of Ca. Particularly in high Ca load cells like the SNc DA neurons, this can lead to very large increases in intracellular Ca. This could alter the membrane potential, particularly by activating SK channels that will hyperpolarize the neurons.This is a possible confound for measuring the contributions of the different inward currents. To learn whether this is a problem, the authors should apply SK blockers to see if the NMDG hyperpolarization is smaller. It might also be advisable to perform Ca imaging (as in Figure 1) to see if the NMDG condition elevates intracellular Ca.

We replied in the above essential revisions.

Other comments:Figure 1C: Presumably the pie chart refers to the durations of the three phases; the legend should say this.

We modified the figure legend.

Figure 1D: Why is the distribution of firing rates here so different from the typical firing rates in (e.g.) Figure 2B?

Most SNc dopamine neurons fire regularly at 1-4 Hz in midbrain slices (Chan et al., Nature, 2007, Jun 28;447(7148):1081-6; Guzman et al., J. Neuroscience, 2009, Sep 2;29(35):11011-9; Hage et al., J. Neurosci., 2012, Feb 22;32(8):2714-21). However, some SNc DA neurons in slice recording show higher firing rates up to 7 Hz transiently or permanently according to many factors, such as exposed to K^+^ internal solution or synaptic excitation, environment, and mice ages (Hage et al., J. Neurosci., 2012, Feb 22;32(8):2714-21; Branch et al., J. Neurosci., 2016, Apr 6;36(14):4026-37). Therefore, in Figure 1D, in order to see the correlation between spontaneous firing rates and phase II slopes, we binned and summed firing rates in a wider range. However, to observe the effects of ion channel blockers on the spontaneous firing rate (requiring a long time for slice recording or acutely isolated neurons), we performed in more stabilized DA neurons regularly firing at 1-4 Hz before we applied ion channel blockers.

Figure 1G: How are these data normalized (i.e. what is the x-axis)?

Figure 1G is an all-point histogram showing the distribution of fluorescence ratio from the trace showing the dendritic ca^2+^ level in Figure 1E. The x-axis shows the counting of fluorescence intensity values (y-axis) from the ROI image during 10 sec recording. In the methods, we described as “Dendritic ca^2+^ changes were presented as a ratio (ΔG/ΔR = (G-G_min_)/(R-R_min_)) after background-subtraction, in which calcium indicator fluorescence (ΔG) was divided by Alexa-594 (ΔR)”.

Figure 1H: The immunostaining for the TRPC3 channel is no more membrane-localized than that for the cytosolic TH enzyme, which seems strange unless most of the channels are in an intracellular pool. The specificity of the antibody should be checked by staining the TRPC3 knockout.

The commercial antibody (Allomone, ACC-016) used for our TRPC3 immunostaining was validated in a previous report (Feng et al., PNAS, 2013, Jul 2;110(27):11011-6) and also in the same mouse (Hartmann et al., Neuron, 2008 Aug 14;59(3):392-8). Our immunostaining shows that TRPC3 fluorescence was observed only in WT, but not in TRPC3 KO mice, too. See Author response image 1.

**Author response image 1. sa2fig1:** Tyrosine hydroxylase and TRPC3 antibody staining in SNc dopamine neurons of wild-type and TRPC3 KO mice. (A, B) Double immunofluorescence staining images for tyrosine hydroxylase (TH, left, red), TRPC3 (TRPC3, right, green), and merge (bottom) from the SNc slices of wild-type (WT, A) and TRPC3 (KO, B) mice..

Figure 1J: the legend says that the figure shows RT-PCR for both TRPC3 and TH, but only one dataset is shown.

We corrected. In fact, we performed this experiment in the TH-eGFP positive cells (that means DA neurons). We made this point clear in the Methods “TH-eGFP positive single dissociated neurons were aspirated into a microelectrode pipette with a sampling solution”.